**RESEARCH**

# Predicting the structural impact of human alternative splicing

Yuxuan Song[1] , Chengxin Zhang[1] , Gilbert S. Omenn[1] , Matthew J. O'Meara[1,2*] and Joshua D. Welch[1,3*] 

*Correspondence:
maom@umich.edu;
welchjd@umich.edu

[1] Department of Computational Medicine and Bioinformatics, University of Michigan, Ann Arbor, MI, USA
[2] Department of Medicinal Chemistry, University of Michigan, Ann Arbor, MI, USA
[3] Department of Computer Science and Engineering, University of Michigan, Ann Arbor, MI, USA

## Abstract

**Background:** Protein structure prediction with neural networks is a powerful new method for linking protein sequence, structure, and function, but structures have generally been predicted for only a single isoform of each gene, neglecting splice variants. To investigate the structural implications of alternative splicing, we use AlphaFold2 to predict the structures of more than 11,000 human isoforms. We employ multiple metrics to identify splicing-induced structural alterations, including template matching score, secondary structure composition, surface charge distribution, radius of gyration, accessibility of post-translational modification sites, and structure-based function prediction.

**Results:** We identify examples of how alternative splicing induces clear changes in each of these properties. Structural similarity between isoforms largely correlates with degree of sequence identity, but we identify a subset of isoforms with low structural similarity despite high sequence similarity. Exon skipping and alternative last exons tend to increase the surface charge and radius of gyration. Splicing also buries or exposes numerous post-translational modification sites, most notably among the isoforms of BAX. Functional prediction identifies numerous functional differences between isoforms of the same gene, with loss of function compared to the reference predominating. Finally, we use single-cell RNA-seq data from the Tabula Sapiens to determine the cell types in which each structure is expressed.

**Conclusions:** Our work represents an important resource for studying the structure and function of splice isoforms across the cell types of the human body.

**Keywords:** Protein structure, Alternative splicing, AlphaFold2, Isoform function, Single-Cell RNA-seq

## Background

Eukaryotic cells achieve remarkable functional diversity from relative compact genomes. This is achieved in part through alternative splicing, where for 90% of genes different combinations of pre-mRNA exons are spliced and ligated together. In humans, although there are approximately 20,000 genes, the number of alternative splicing events can reach up to 100,000 [1, 2, 3]. Alternative splicing can exert diverse biological effects on

proteins, including alterations in their binding properties, subcellular localization, and stability [4]. Moreover, aberrant alternative splicing can disrupt normal biological processes and lead to Duchenne muscular dystrophy [5], cardiovascular disease, and multiple types of cancers [6]. Alternative splicing has been extensively studied from the perspectives of regulation [7], evolution [8], and expression profile [9]. In contrast, the impact of alternative splicing on protein structure has been studied much less. Previous studies have shown that alternative splicing can induce unstable protein conformations [10], change protein localization [4], alter transmembrane domains [11], and create variations in repeat regions [12, 13]. However, these studies have been relatively constrained in scope [13], and a thorough and systematic investigation of how splicing affects structure is imperative.

For decades, obtaining experimental protein structures has been a laborious and time-consuming process. As a result, structures of multiple spliced isoforms from the same gene have not often been experimentally solved. Computational protein structure prediction methods like Rosetta and I-TASSER offer a more practical means of studying isoform structures on a large scale [14, 15]. Most recently, neural network approaches such as AlphaFold2 and RoseTTAFold have enabled high-precision protein structure prediction [16, 17]. These tools provide an exciting new opportunity to study how alternative splicing affects protein structure. Recently, Sommer et al. utilized ColabFold to predict structures for more than 127,000 human spliced isoforms annotated from RNA-seq experiments [18, 19]. They used protein structure confidence scores (pLDDT) to nominate a reference isoform for each gene based on the assumption that predicted structures for loss-of-function isoforms will have lower confidence. However, Sommer et al. did not investigate the structural differences among isoforms any further, and the details of how splicing changes specific structural properties of proteins remain unexplored. Additionally, the functional implications of such structural changes remain unclear.

Another important question concerns the cell-type-specific expression of isoforms. Single-cell RNA sequencing (scRNA-seq) has enabled us to map the expression profiles of individual genes and isoforms at cellular resolution [20]. Prior investigations utilizing scRNA-seq have delineated differential isoform usage in contexts such as the adult brain [21] and muscle cell maturation [22]. Multiple studies have profiled nearly the entire human body with scRNA-seq [23–25], raising the exciting possibility of mapping the cells within which each predicted structure is present. This would be an important step toward understanding how alternative splicing changes the functions of proteins within different cellular contexts.

In this study, we addressed these questions by using AlphaFold2 to predict structures of more than 11,000 human splice isoforms annotated in UniProt. We then analyzed the structures of our isoforms plus the 127,000 isoforms folded by Sommer et al. using a variety of metrics to identify splicing-induced structural changes [19]. We identified numerous ways in which splicing affected structure, including changes in secondary structure, surface charge, protein compactness, and the surrounding environment of post-translational modification (PTM) sites. Additionally, we integrated the predicted structural information with expression data from the Tabula Sapiens [23], which includes scRNA-seq experiments spanning the whole human body. Finally, we used structure-based function prediction to evaluate the functional consequences of alternative splicing. Our

workflow is summarized in Fig. 1 with examples of each alternative splicing (AS) type in Additional File 1: Fig. S1A. While AlphaFold2 has high predictive accuracy and well calibrated confidence estimates, a limitation of this computational analysis is that the

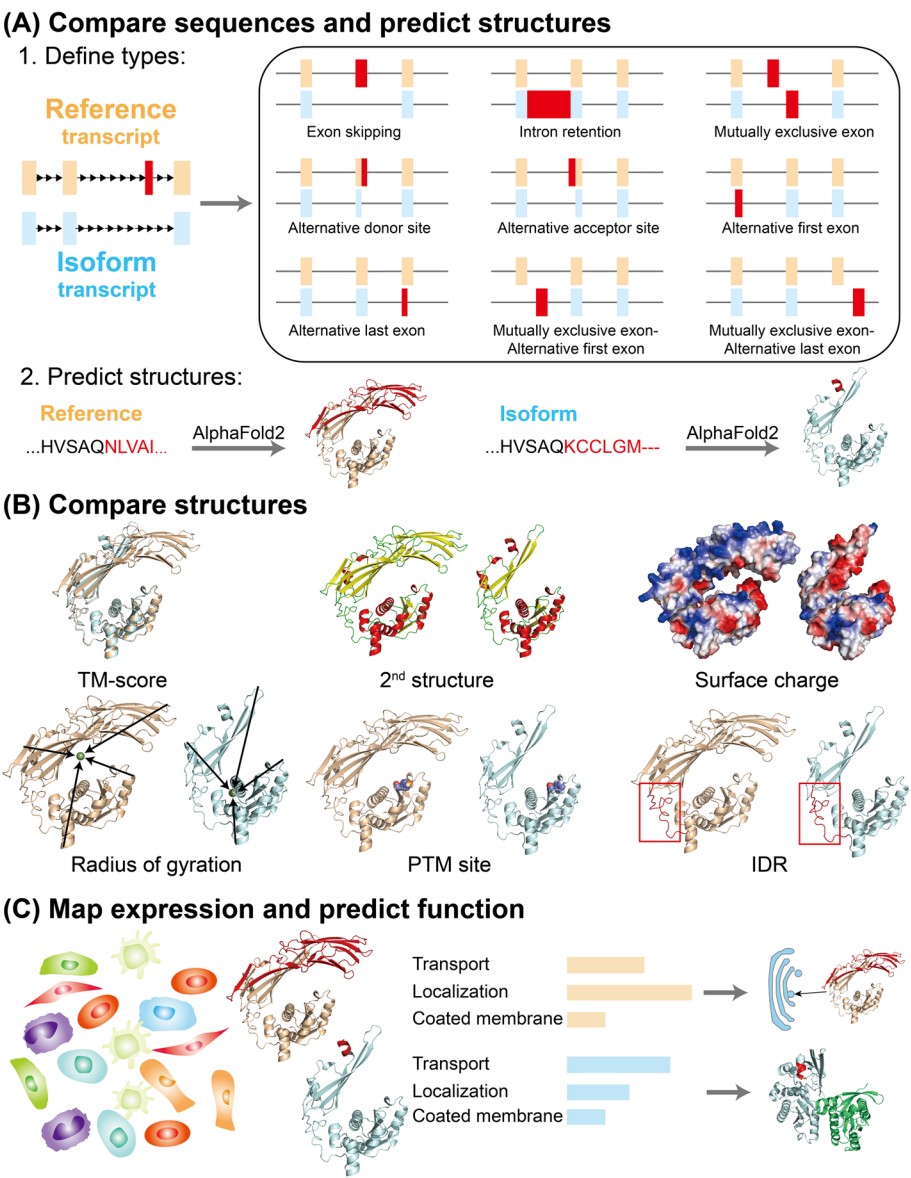

**Fig. 1** Workflow for exploring the structural effects of alternative splicing. The workflow is composed of three parts: **A** Compare sequences and predict structures: we use AlphaFold2 to predict the structure of 11,161 human alternative spliced isoforms from 5966 genes. The predicted structures for the "reference" isoform of each gene are publicly available. For each isoform, we annotate the alternative splicing type based on the pattern of spliced exons relative to the reference isoform, and in total we identify nine alternative splicing types among 7923 isoforms. **B** Compare structures: we compare the structures of isoforms for each gene using six different metrics. We calculate template-matching score (TM-score), secondary structure percentage, surface charge, radius of gyration, solvent-accessible area of post translational modification (PTM) sites and intrinsically disordered region locations. **C** Map expression and predict function: We quantify the expression for each isoform from the Tabula Sapiens scRNA-seq dataset using Kallisto and identify isoform expression differences across human cell types. We use COFACTOR to predict protein functions based on their structures and compare the predicted gene ontology (GO) terms for reference and isoform transcripts

predicted structure may be inaccurate and that inaccuracies may be correlated with alternative splicing, e.g., through the differences in sequence homology captured by the input multiple-sequence alignments, or if the structural properties are qualitatively different in splice variants. To test for the sensitivity of the analysis to these factors, we also re-analyzed the impact of splicing on structural features using stringent filtering for predicted intrinsically disordered regions and isolated helices. An additional limitation of this study is that we focus on single-chain prediction, while many proteins have post-translational modifications or function in obligate or transient complexes. We anticipate this work will be a useful baseline measuring the impact of improvements in structure prediction methods going forwards.

## Results

### Assessment of AlphaFold2 structure prediction for human alternatively spliced isoforms

Recent studies have successfully predicted the three-dimensional structures of the human proteome [26, 27]. These works capture only one structure per gene, however, neglecting the complexity of alternative splicing where multiple isoforms can be expressed. Recently, Sommer et al. predicted the structures for over 127,000 human transcripts from the CHESS database of gene annotations. The goal of the Sommer study was to use predicted structures to nominate a canonical or "reference" isoform of each gene by identifying well-folded gene products [19, 28]. A key question left unanswered in that work was: What is the overall structural impact of human alternative splicing? To begin to answer this question, we identified all human splice isoforms in SwissProt. After filtering to a maximum of 600 amino acids and removing the "reference" isoforms which have already been folded, we obtained 11,159 isoforms. We then compared our predicted isoform structures with the publicly available structures for the 5966 isoforms annotated as "reference" isoforms in the AlphaFold Protein Structure Database (https://alphafold.ebi.ac.uk/). The advantage of using isoforms from SwissProt is that many annotations are available, such as the locations of post-translational modification sites. We also incorporated the structures predicted by Sommer et al. in our analyses (see below).

While the structures predicted by AlphaFold2 are generally high-quality with well-calibrated confidence scores [29], AlphaFold2 predictions remain inaccurate in certain contexts. For example, AlphaFold-Multimer and AlphaMissense were developed to address limitations in the prediction of protein complexes and the structural impact of point mutations [30–32]. A key insight from these and other works is that the quality of the structure prediction in deep-learning-based structure prediction methods like AlphaFold2 depends on the abundance and quality of multiple sequence alignments (MSAs) for the targets to be folded [33]. Because alternative splicing is more prevalent in higher eukaryotes [34], the conservation across variable exons in alternatively spliced isoforms may vary. This variation in MSA depth could influence the quality of the predicted isoform structures and potentially confound overall interpretation of the structural impact of alternative splicing.

We thus investigated how variability in MSA information affects the prediction of structures for alternatively spliced transcripts. To do this, we extracted the MSA depth for each residue across the isoforms we folded and mapped the different alternative splicing regions marked as "replaced" or "missing" and non-AS regions with their MSA

counts (Fig. 2A). "Missing" denotes sequences found in the reference isoform but absent in the alternate isoform. "Reference replaced" describes sequences present in the reference isoform that are substituted by new regions in the alternate isoform. "Isoform replaced" identifies sequences unique to the alternate isoform, not present in the reference. "Isoform other" and "Reference other" indicate regions shared by both the reference and alternate isoforms. A diagram explaining the differences among these types of splicing is shown in Additional File 1:Fig. S1B. As expected, non-AS regions exhibited a higher MSA depth compared to AS regions (analysis of variance (ANOVA) *p*-value: 0.0). We also compared the MSA depth of exons that were missing from the reference

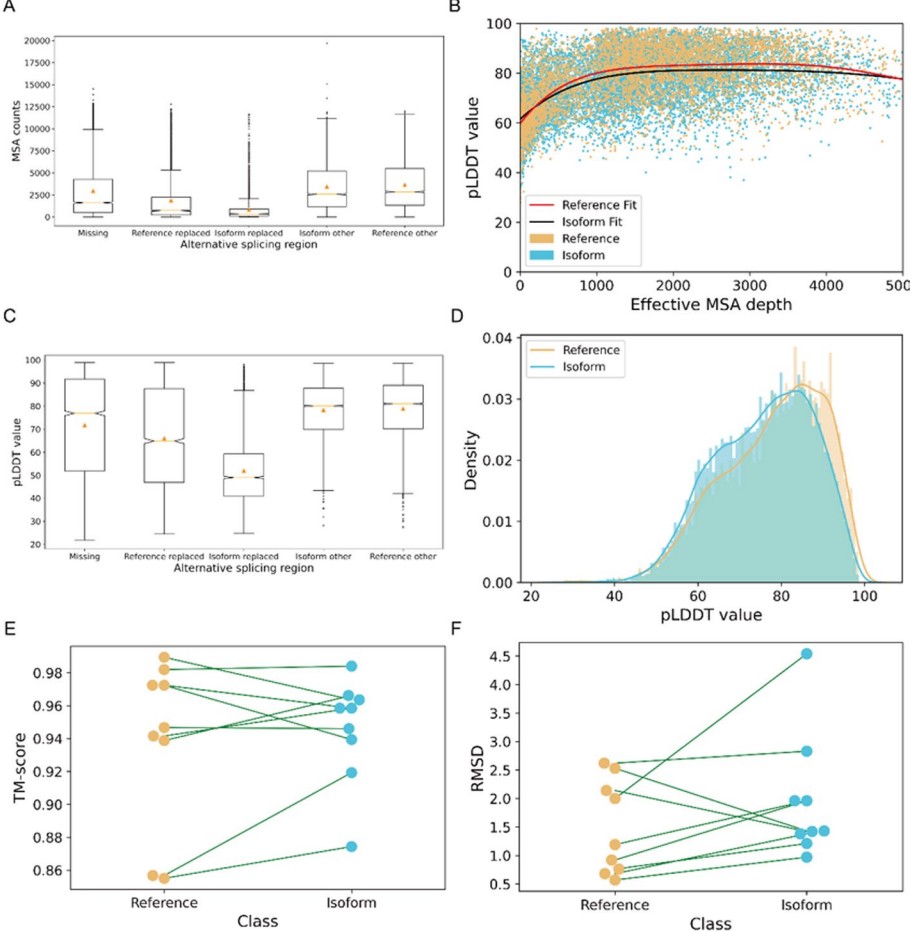

**Fig. 2** Reliability assessment of AlphaFold2 predictions for human alternatively spliced isoforms. **A** Box plots (orange triangle: mean, box: 25–75% quantile range, dots: outlier values) of MSA counts across alternatively spliced vs. constitutive regions. **B** Scatter plot of pLDDT value by effective MSA depth for reference and alternative isoform structures. Polyfit trend lines for reference and isoform are colored in black and red. **C** Box plots of pLDDT values across alternatively spliced vs. constitutive regions. ANOVA shows that differences in mean MSA counts and pLDDT among missing, reference replaced, isoform replaced, isoform other and reference other regions are significant (*p*-value < 1.33e -301 for MSA and *p*-value < 5.04e-295 for pLDDT). **D** Density plot of prediction quality for 5966 reference and 11,161 isoform structures predicted by AlphaFold2 from sequences annotated in SwissProt. Distribution of prediction quality for CHESS dataset refers to Additional File 1:Fig. S2A. There are 9 genes that have experimental structures with resolution greater than 3 Å deposited in Protein Data Bank (PDB) for both the reference and an alternate isoform. For these pairs, **E** shows the TM-score (paired *t*-test *p*-values: 0.76) and **F** shows the RMSD (paired *t*-test *p*-values: 0.32) comparing the AlphaFold2 predicted structures against the ground truth experimental structures

("missing") or replaced with a difference sequence ("replaced"). The replaced regions in the isoforms displayed the lowest MSA counts, lower than the missing regions, which in turn had fewer MSA counts than the regions that were shared between reference and alternate isoforms ("reference other" and "isoform other"). The structure confidence score (predicted local distance difference test, pLDDT) values for isoform structures reflected the differences in MSA depth, with replaced regions having the lowest pLDDT, missing regions slightly higher, and unchanged regions having the highest pLDDT. However, some structures in each category were folded with high confidence. Importantly, the pLDDT scores depended on the MSA depth in a similar way regardless of whether the sequence was a reference or alternate isoform (Fig. 2B, Pearson correlation coefficients (PCC) of 0.475 for reference isoforms and 0.529 for alternate isoforms). Consequently, the AS regions demonstrated lower structural quality compared to non-AS regions (Fig. 2C). While the overall distribution of pLDDT scores was broadly similar, the average pLDDT was slightly lower for predicted isoform structures compared with reference structures (Fig. 2D). This trend was also observed by Sommer et al. when they folded isoforms from the CHESS database [19] (Additional File 1:Fig. S2A). In summary, we find that MSA depth does influence the structural prediction quality of AS regions, but pLDDT reflects this dependence for both reference and alternate isoforms. Thus, it should be possible to use pLDDT to filter out isoforms whose structures cannot be accurately predicted due to low MSA depth.

To further investigate the accuracy of AlphaFold2 predictions for isoform structures, we searched the Protein Data Bank (PDB) database for proteins with experimentally determined structures for multiple isoforms. We identified 11 such proteins with experimental structures for at least 2 distinct isoforms and for which the structures include the alternatively spliced region (Additional file 2:Table S1). Of the 22 experimentally determined structures meeting these criteria, 20 were resolved using X-ray crystallography, one (UBE2V1) was determined using Nuclear Magnetic Resonance spectroscopy, and one (ELOC) was determined by Electron Microscopy. We excluded ELOC and UBE2V1 from the comparison because of the low resolution (8.20 Å) of the ELOC isoform structure and lack of resolution information of the UBE2V1 structure. The other structures determined by X-ray crystallography have resolutions ranging from 1.45 to 3.72 Å. Reassuringly, the AlphaFold2 predictions matched the experimentally determined structures equally well for both reference and alternate isoforms (Fig. 2E-F). The template-matching score (TM-score) and root mean squared deviation (RMSD) for each experimental-predicted structure pair were not significantly different between isoform and reference structures, with paired *T*-test *p*-values of 0.76 and 0.32 (Fig. 2E,F). We note that AlphaFold2 was trained on the PDB, likely including these structures. We folded these sequences using a template-free version of AlphaFold2 to ensure that the PDB structures were not directly used in the prediction. It is possible that the results of our comparison are overly optimistic in assessing the quality of isoform structure prediction because the structures were in the AlphaFold2 training dataset. However, the analyses in the AlphaFold2 paper indicate that the model is robust to small changes in training data [16], so these predictions would likely still be similar if AlphaFold2 were re-trained without the structures included. Overall, we take this as promising but not definitive evidence that AlphaFold2 can accurately predict isoform structures.

Given the broad overall similarity in structural quality between reference and alternate isoforms and trusting the pLDDT as a measure of structural quality, in the next sections we proceed to explore the structural differences between 4450 reference and 7631 isoform structures with high and confident prediction quality (pLDDT $\geq$ 70), while being aware of the limitation of the analysis that differences in MSA depth in alternatively spliced regions may lead to lower structural quality in some cases.

Intrinsically disordered regions (IDRs) and isolated alpha helices are two other potential confounders in comparing predicted structures. We observed that AlphaFold2 occasionally predicts structures containing extended alpha helices possessing limited interactions with the rest of the protein. These helical regions often receive high pLDDT scores despite being uncommon in experimentally determined structures. In addition, IDRs are highly flexible in real proteins, but AlphaFold2 predicts a single conformation for them. Thus, IDRs can strongly affect some structural metrics like radius of gyration, masking changes in protein size and compactness, which could bias our conclusions. Therefore, we established a set of filtering criteria to construct subsets of our dataset by excluding structures containing isolated helices or intrinsically disordered regions (IDRs). We combined residue-level relative solvent accessibility (RSA) with secondary structure type to identify isolated helices in the predicted structures (see Methods). For IDRs, we consider both sequence and structure-based definitions of IDRs (see Methods), but favor structure-based criteria due to the high-predictive power recently described in Tesei et al. [35]. Applying these filters, we curated a dataset of 975 reference structures and 1359 isoform structures, excluding both IDRs and isolated helices. We performed parallel analyses on both the whole dataset (including structures with significant IDRs and/or isolated helices) and the subsets filtered to exclude structures with IDRs and/or isolated helices (Additional File 1: Fig. S3).

### Structural similarity of isoforms generally reflects sequence similarity, but some structures diverge despite similar sequences

A sensible initial hypothesis about the effects of alternative splicing on protein structure is that removing or replacing large parts of the reference sequence should have large effects on the structure. Conversely, mutually exclusive exons with similar size or small changes in exon boundaries should generally result in smaller structural changes. To explore these effects, we examined the relationship between the sequence length of the alternatively spliced region and the template matching score (TM-score) for each isoform compared to the reference. We annotated each of the splicing events in our set of isoforms using the following splicing events: exon skipping (ES), alternative donor site (ADS), alternative acceptor site (AAS), mutually exclusive exons (MXE), intron retention (IR), alternative first exon (AFE), alternative last exon (ALE), mutually exclusive exon-alternative first exon (MXE-AFE) and mutually exclusive exon-alternative last exon (MXE-ALE) (Fig. 1). Note that multiple types of splicing events may occur in a given isoform. To quantify the degree of sequence change, we calculated the length of each splicing event. Our findings indicate a negative correlation between the TM-score and length of alternatively spliced sequence across all nine types of alternative splicing, with the strongest relationship for MXE-ALE, with a PCC of -0.547 (Fig. 3A), and we observe similar results in the non-IDR, non-isolated helices dataset, with the strongest result in

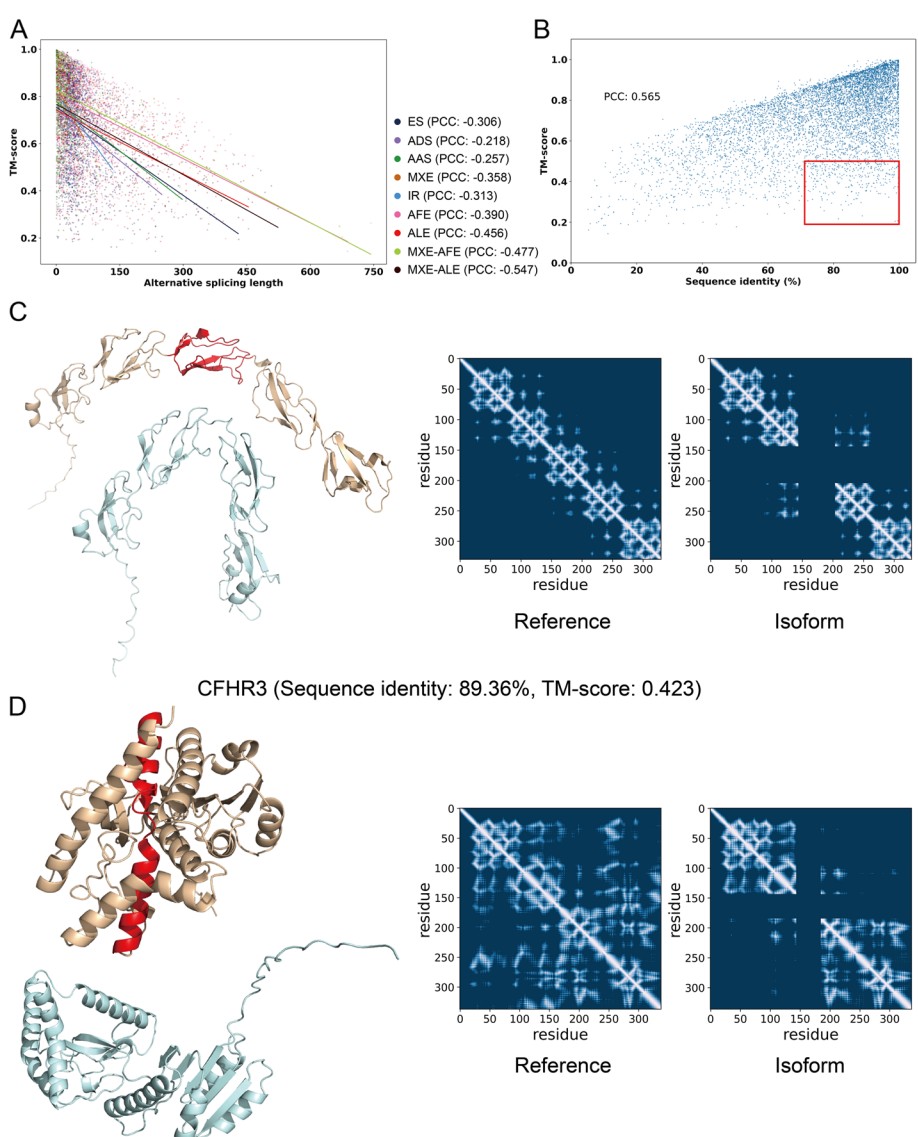

**Fig. 3** Structural similarity of isoforms generally reflects sequence similarity, but some structures differ despite similar sequences. **A** Scatter plot of length of alternative splicing region (*x*-axis) vs. TM-score between reference and alternate isoform structure (*y*-axis) colored by alternative splicing type. Fitted lines are presented for each alternative splicing type. The same analysis was performed on the non-IDR, non-isolated helices dataset, as referred to in Additional File 1:Fig. S4A. **B** Scatter plot of percent sequence identity between reference and alternate isoform (*x*-axis) vs. TM-score (*y*-axis) (PCC = 0.565). The high sequence identity (> 70%) but low TM-score (< 0.5) isoforms are shown in the red box. The same analysis for the CHESS dataset and the non-IDR, non-isolated helices datasets are shown in Additional File 1:Fig. S2B and Additional File 1:Fig. S4B. Examples of isoforms with high sequence identity but low TM-score: CFHR3 (**C**) and MDH2 (**D**). The reference and isoform structures are colored in wheat and pale cyan, and the alternatively spliced regions are colored in red. The contact maps for references and isoforms are aligned to the same length. The isoforms predominantly influenced by structural irregularities in loop regions are illustrated in Additional File 1: Fig. S7. The observed variability in the loop regions may arise either from inherent structural disorder or as a consequence of AlphaFold2 predictions characterized by lower accuracy

IR (PCC: − 0.681) (Additional File 1: Fig. S4A). This indicates that, as expected, larger sequence changes due to splicing tend to cause larger structural changes.

We additionally calculated percent sequence identity based on sequence alignment between isoform and reference. We observed a positive correlation between sequence identity and TM-score (PCC: 0.565) (Fig. 3B). This indicates again that the isoforms of a gene generally have more similar structures when they share a higher sequence identity. We confirmed a similar relationship in the isoforms of the CHESS dataset (PCC: 0.783) (Additional File 1: Fig. S2B), and in the non-IDR, non-isolated helices dataset (PCC: 0.798) (Additional File 1: Fig. S4B). Some isoforms can have very low sequence identity compared to the reference isoform (< 30%), such as truncated isoforms that are much shorter compared to their references. In such cases, the TM-score between reference and alternate isoform structure is always correspondingly low. In many cases, low sequence identity between reference and alternate isoforms results from the loss of a large portion of amino acids. One example is EEF1A lysine methyltransferase 3 (EEF1AKMT3), which mediates the methylation of eukaryotic elongation factor 1 alpha (eEF1A) at Lys-165 [36]. Its isoform shares a sequence identity of 43.43%, with portions of the alternative splicing regions retained in the isoform (Additional File 1:Fig. S5A top). According to UniProt annotations, there are seven S-adenosyl-L-methionine binding sites in the reference structure, four of which are preserved in the isoform (Additional File 1: Fig. S5A bottom). Due to ES and ALE events in the isoform, residues 104 W and 133 W in the reference isoform are replaced by 104E and 133P in the alternate isoform. These changes alter the spatial arrangement of the binding sites, potentially impairing its ability to bind S-adenosyl-L-methionine. We also observed cases where the isoform and reference sequences are similar in length but share low sequence identity, yet still exhibit regions of structural similarity. For instance, signal recognition particle 9 kDa protein (SRP9) is a component of the recognition particle (SRP) complex, and has an isoform with a sequence identity of 47.98%. Despite the low sequence identity, the alternative splicing regions are structurally similar between the isoform and reference (Additional File 1: Fig. S5B top), resulting in a high TM-score of 0.76. The alternative splicing regions may preserve similar interactions, as suggested by the side chain orientations for each residue between the reference and isoform structures (Additional File 1: Fig. S5B bottom).

However, this analysis highlighted a subset of isoforms that have low structural similarity despite high sequence similarity. These examples are particularly interesting to examine in our study, because the structural effects of alternative splicing cannot be predicted from the length of the spliced portion of the isoform and thus require structures to detect. Three hundred twenty eight isoforms were characterized by high sequence identity (> 70%) but low TM-score (< 0.5) (Additional file 3:Table S2). Among our high-quality structures, we identified 53 isoforms where the structured domains were altered owing to alternative splicing, and thus may have functional implications. We noticed that the structural differences for the 53 isoforms largely fall into three groups: different helix orientation (CENPX, SNAP23, IL32, etc.), loss of repeat domain (SIGLEC8, CFHR3, BTN3A1, etc.), and change in structural compactness (LIAS, NDRG2, CHAC1, etc.) (Additional File 1: Fig. S6), even though some examples could be low prediction from the AlphaFold2, suggested by the long IDRs and isolated helices. For example, the CFHR3 (Complement factor H-related protein 3) reference isoform contains five repeated Suchi

domains, which mediate cytokine binding [37]. From the structure and the residue contact map, CFHR3 isoform 2 splices out one Suchi domain (Fig. 3C). Similarly, in mitochondrial malate dehydrogenase (encoded by MDH2), the isoform 2 spliced out a region from residue 144 to residue 185, a structural region of helices, resulting in a noticeably looser conformation (Fig. 3D). This ES event in the MDH2 isoform 2 also causes the loss of critical malate binding site 176R [38], and therefore may impair substrate binding in the isoform. Because loops can be highly flexible, their precise structures are difficult to predict computationally, and thus it is difficult to draw many conclusions about them. Gene ontology (GO) enrichment analysis for the genes associated with the other 275 isoforms suggests that their functions can be broadly categorized into immune response (CD244, CD1E, CD276, etc.) and protein transportation (CTLA, SYT4, SYT15, etc.) (Additional File 1:Fig. S7). For example, the structures of the immune response receptors are presented in a similar pattern: an Ig-like extracellular domain, a single-helix transmembrane domain followed by an unstructured cytoplasmic region; variation in the linker regions or the cytoplasmic region could result in different orientation of domains which will lead to a low TM-score in the structural alignment.

### Alternative splicing changes secondary structure, surface charge, and radius of gyration

Our predicted structures allow us to investigate structural changes that are not obvious from sequence changes but have important implications for protein function. To investigate how splicing may change such properties, we calculated the surface charge and radius of gyration for our predicted structures and quantified differences in these metrics between reference and alternate isoforms. We also investigated how splicing changed secondary structures (helix, sheet, and loop) by comparing the percentage of each secondary structure for each alternative splicing type (Additional File 1:Fig. S8A). For most alternative splicing types, the distribution of secondary structure percentage between isoform and reference is similar; however, exceptions occurred at the beginning and end of proteins. For example, alternative first exon events tend to create more helix and less loop in the isoform. This phenomenon is even more obvious in the CHESS dataset (Additional File 1:Fig. S8B).

In contrast to structural similarity and secondary structure percentage, which predominantly influence protein stability, structural attributes such as surface charge and radius of gyration may carry greater significance in shaping alterations in protein function. For example, the location of charged residues is crucial for electrostatic interactions in which proteins participate. Surface charge thus plays a pivotal role in various processes, including protein ion binding and protein localization [39, 40]. Additionally, the overall compactness or looseness of a structure can dramatically alter protein functions such as transport or catalysis. The radius of gyration serves as a metric for evaluating protein compactness and may serve as an indicator of potential intrinsically disordered regions (IDRs) [41, 42].

We calculated the total surface charge by first identifying surface residues based on the relative solvent accessibility (RSA) for each residue of the structure, then summing the charges of the surface residues. The overall distribution of surface charge does not show significant difference between reference and alternate isoforms (Mann–Whitney *U* test *p*-value: 0.304) (Fig. 4A), with a median of 0.99 in the reference group and 1.00

in the isoform group. We observed the same result in the non-IDR, non-isolated helices dataset (*p*-value: 0.577) (Additional File 1:Fig. S4C). However, if we compute the difference in surface charge between each isoform structure and its corresponding reference structure, we observed outliers where the isoform surface charge increased or decreased significantly (Fig. 4B). We observed 231 positive surface charge outlier isoforms, where alternate isoforms have from 9 to 31 more units of surface charge than the reference isoforms. There are 214 negative surface charge outlier isoforms, where isoforms have between −56 and −9 units of surface charge compared to the references. The five most frequent alternative splicing types among isoforms with significant surface charge changes are ES, ALE, AFE, MXE-ALE, and MXE-AFE in the high and confident dataset (Fig. 4C), and ES, ALE, AFE, MXE-ALE, and ADS in the non-IDR, non-isolated helices dataset (Additional File 1:Fig. S4E). Interestingly, positive and negative surface charge changes are about equally likely to occur for AFE, MXE-ALE, and MXE-AFE, but ES and ALE more often increase surface charge. A notable example among these positive outlier isoforms is C-X-C chemokine receptor type 3 (CXCR3). An AAS event in isoform 2 causes a 47 residue insertion in the N-terminal portion of the protein; the extended loop inserts into the binding pocket of CXCR3 and buries 99D, 159D, and 325D, which are all positively charged aspartic acid residues (Fig. 4D top). In addition, this insertion causes a much more positively charged surface in CXCR3 isoform 2, which could explain the binding preference between the reference and the isoform, where the ligands (CXCL9, CXCL10, and CXCL11) for the reference have much a more positively charged "head" than the ligand (CXCL4) binding for the isoform [43] (Fig. 4D bottom). RNA 3'-terminal phosphate cyclase-like protein (RCL1), which serves a role in ribosome biogenesis, is an interesting example from among the negative outlier isoforms. An AFE event in the isoform causes a loss of 186 residues at the N-terminal, which exposes negatively charged residues including 99E, 100E, and 108D, creating a more negatively charged surface in the isoform (Fig. 4E).

Since intrinsically disordered regions strongly affect the radius of gyration, we consider changes in radius of gyration only for those where both the reference and isoform are predicted to not have intrinsically disordered regions within the SwissProt dataset with high and confident prediction quality and conduct the relevant statistical analysis.

The overall distribution of radius of gyration is similar between reference and alternate isoforms, but the median radius is slightly lower for alternate isoforms in the high and confident dataset (22.9 Å vs 22.0 Å) and the non-IDR, non-isolated helices dataset (22.2 Å vs 21.1 Å) (Fig. 5A and Additional File 1:Fig. S4F). The distribution of the difference of radius of gyration (isoform radius minus reference radius) is negatively skewed, with a median of −0.9 Å in the high and confident dataset and −0.5 Å in the non-IDR, non-isolated helices dataset (Fig. 5B and Additional File 1:Fig. S4G). This trend likely reflects a general loss of sequence and shorter overall protein length in alternate isoforms relative to the reference isoform. The five most frequent alternative splicing types among isoforms that are outliers in terms of radius of gyration changes are ES, AFE, ALE, MXE-ALE, and AAS, and there are more negative outliers than positive outliers (Fig. 5C and Additional File 1:Fig. S4H). ES and ALE are especially likely to decrease the radius of gyration.

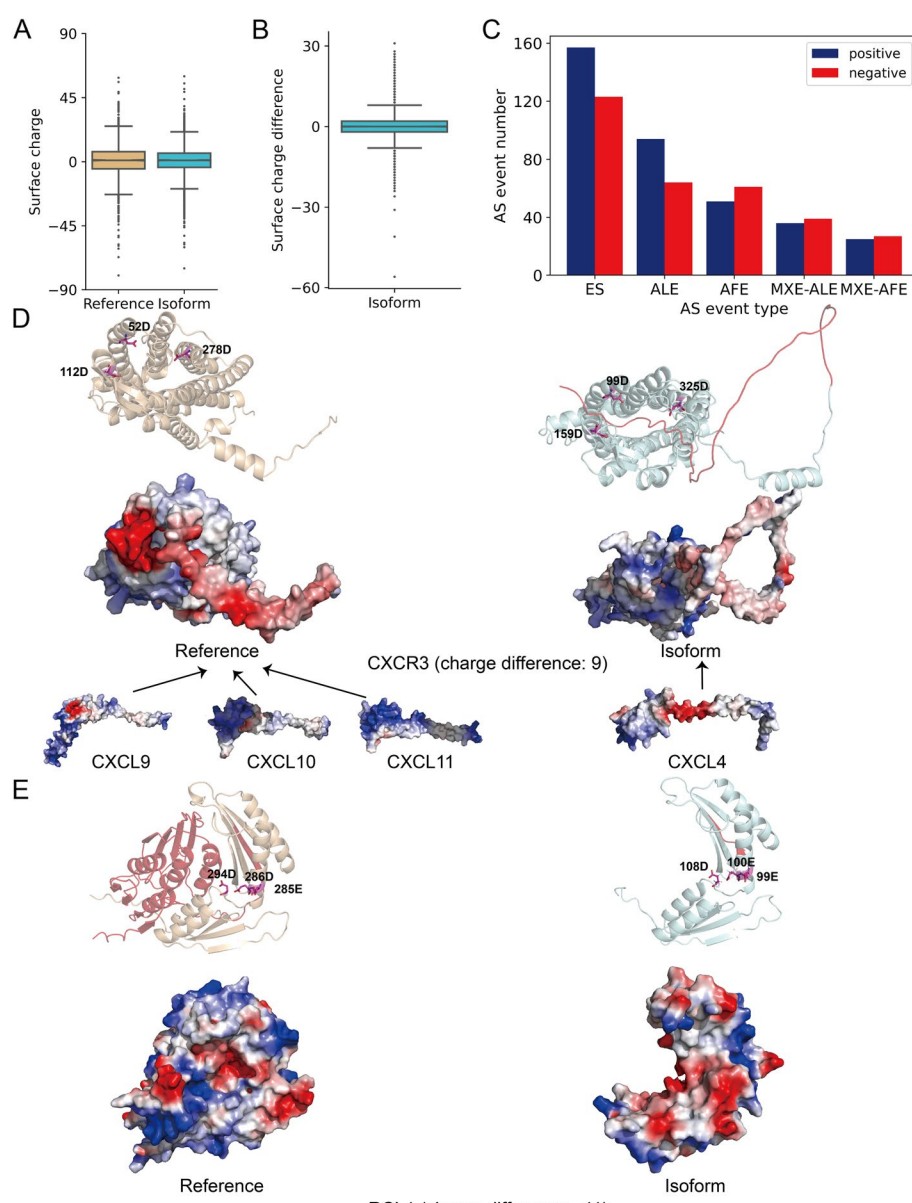

**Fig. 4** Alternative splicing changes surface charge. **A** Overall surface charge distribution between reference and isoform structures for SwissProt dataset with high and confident prediction quality (*p*-value: 0.304, Mann–Whitney *U* test), with outliers (defined as Q1 − 1.5 IQR and Q3 + 1.5 IQR) occurring at both ends, $N = 7024$. Surface charge distribution for the CHESS dataset and non-IDR, non-isolated helices dataset are shown in Additional File 1: Fig. S2C and Additional File 1: Fig. S4C. **B** Differences of surface charge for the SwissProt dataset with high and confident prediction quality. Difference of surface charge is calculated by using the isoform surface charge minus the reference surface charge, $N = 4952$. The difference of surface charge for the CHESS dataset and the non-IDR, non-isolated helices dataset are shown in Additional File 1:Fig. S2D and Additional File 1: Fig. S4D. **C** The five most frequent alternative splicing events in the positive and negative surface charge outliers. The results for the CHESS and the non-IDR, non-isolated helices datasets are presented in Additional File 1: Fig. S2E and Additional File 1: Fig. S4E. **D** Example of positive surface charge outliers: CXCR3. The reference isoform binds different ligands (CXCL9, CXCL10, and CXCL11) compared to the ligand that binds the alternative isoform (CXCL4). The three aspartic acid residues are labeled in pink. Alternatively spliced portions of the structure are colored red. **E** Example of negative surface charge outliers: RCL1, acidic amino acids are labeled in pink sticks

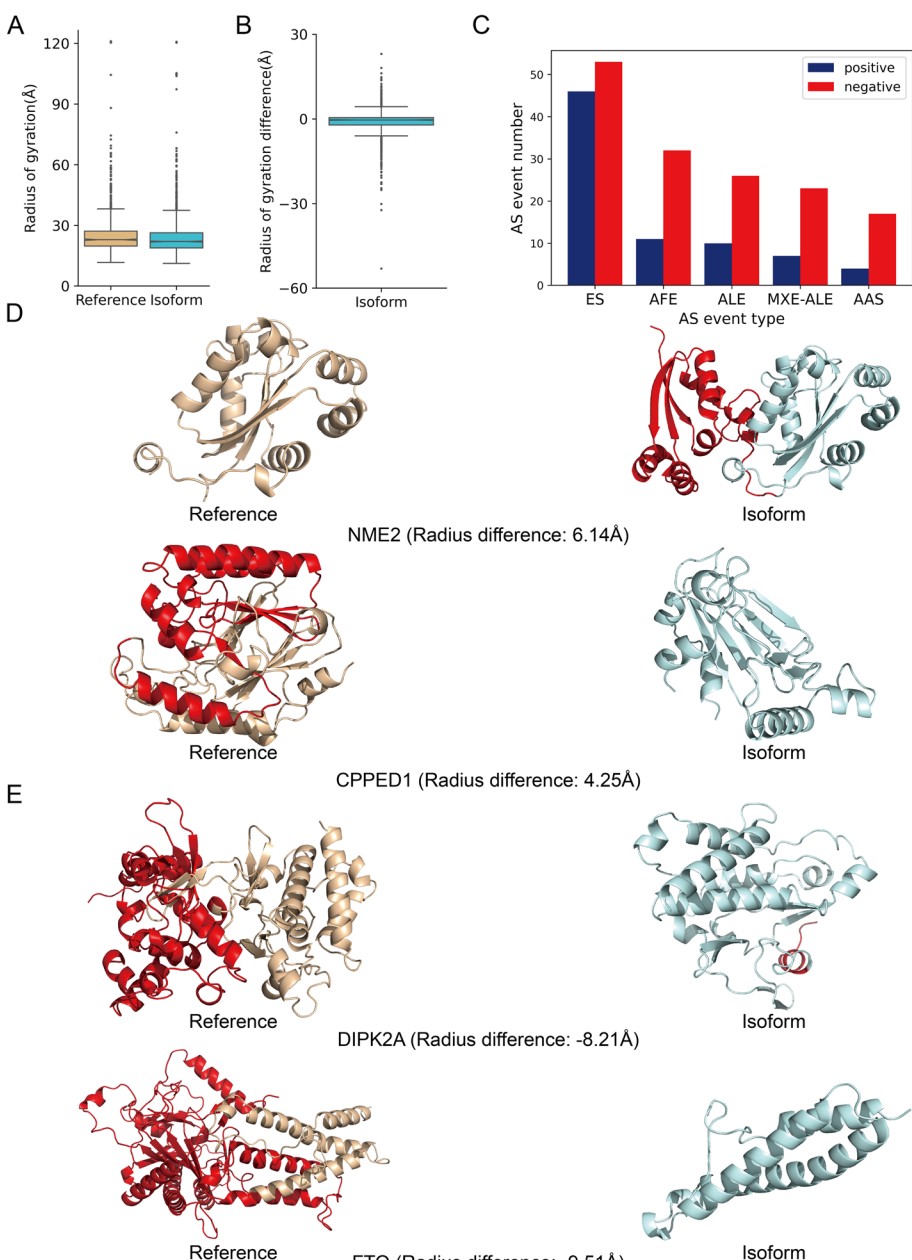

**Fig. 5** Alternative splicing changes protein compactness as measured by radius of gyration. **A** Box plot of radius of gyration values for reference and alternate isoform structures (*p*-value: 1.623e-5, Mann–Whitney *U* test), $N = 1770$. The median difference between reference and alternate isoform radius is $-0.9$ Å, the outliers defined as Q1 − 1.5 IQR and Q3 + 1.5 IQR. Radius of gyration distribution for the CHESS and the non-IDR, non-isolated helices datasets are shown in Additional File 1:Fig. S2F and Additional File 1:Fig. S4F. **B** Differences in radius of gyration for the SwissProt dataset, $N = 1328$. Results for the CHESS and the non-IDR, non-isolated helices dataset are shown in Additional File 1: Fig. S2G and Additional File 1: Fig. S4G. **C** The five most frequent alternative splicing events in positive and negative outliers. Results for the CHESS and the non-IDR, non-isolated helices dataset are shown in Additional File 1:Fig. S2H and Additional File 1: Fig. S4H. **D** Examples of positive outliers of radius of gyration (NME2 and CPPED1). Alternatively spliced portions of the structure are colored red. **E** Examples of negative outliers of radius of gyration (DIPK2A and FTO)

In some cases, extensions in the isoform can lead to a higher radius of gyration. For example, the isoform of nucleoside diphosphate kinase B (NME2), produced from a readthrough transcript of NME1 and NME2 [44], contains an extended region at the N-terminal (Fig. 5D top). This extension results in an estimated 6.14 Å increase in the radius of gyration. Given that NME1 and NME2 usually form hexamers, and the NME2 isoform incorporates regions from both proteins, it is possible that the NME2 isoform may regulate the formation of NME1-NME2 hexamer. Similarly, a positive radius of gyration can result from skipping events. For example, Serine/threonine-protein phosphatase CPPED1 (CPPED1) is involved in regulating signaling pathways by dephosphorylating Akt family proteins [45]. An ES event in its isoform removes regions composed of helices and sheets, resulting in a looser conformation with an increased radius of gyration (Fig. 5D, bottom). In contrast, negative outliers primarily result from the loss of structured regions. Divergent Protein Kinase Domain 2 A (DIPK2A) is a protein involved in the regulation of autophagy [46]. Its isoform loses 209 residues due to an MXE-AFE event, but the kinase domain is largely retained (Fig. 5E top). Similarly, Alpha-Ketoglutarate-Dependent Dioxygenase (FTO) is an mRNA demethylase studied for its role in the regulation of obesity [47]. An AFE event in its isoform results in the loss of 399 residues, including the N-terminal domain (NTD) responsible for its demethylase activity (Fig. 5E bottom).

To summarize the relationship between alternative splicing types and structural properties examined so far, we performed a regression analysis to predict the values of TM-score, surface charge difference, secondary structure percentage difference, and radius of gyration difference as a function of the number of residues changed by each alternative splicing type. This regression model provides a principled way to control for confounding factors, including total protein length and structure quality. The effect size represents the per-residue effect of each alternative splicing type on TM-score, secondary structure percentage (helix, sheet, loop), surface charge and radius of gyration.

The regression analysis indicates that most alternative splicing types induce a statistically significant decrease in TM-score, except for MXE, for the high and confident SwissProt dataset (Fig. 6A). This could be because MXE often results in relatively small sequence changes. Furthermore, ES, ALE, and MXE-ALE stand out as the three alternative splicing events exhibiting the most pronounced and statistically significant negative per-residue impact on the TM-score. Alternative splicing has little per-residue effect on the secondary structure percentage; only splicing events at the beginning and end of the protein significantly affect the secondary structure (Fig. 6A). Interestingly, intron retention in the alternate isoform shows a significant positive effect on surface charge. We also found that alternative splicing types leading to loss of residues, like exon skipping, exert a positive per-residue influence on the difference of radius of gyration. This makes sense, because isoforms with fewer residues will generally have a lower radius of gyration. Since different metrics, particularly the radius of gyration, can be influenced by intrinsically disordered regions, we also performed regression analysis on the non-IDR, non-isolated helices dataset comprising 1359 splicing isoforms (Fig. 6B). The overall results were consistent with the original analysis: MXE was the only alternative splicing type that showed no significant effect on the TM-score, while other splicing types were associated with a decrease in TM-score. Additionally, most alternative splicing types

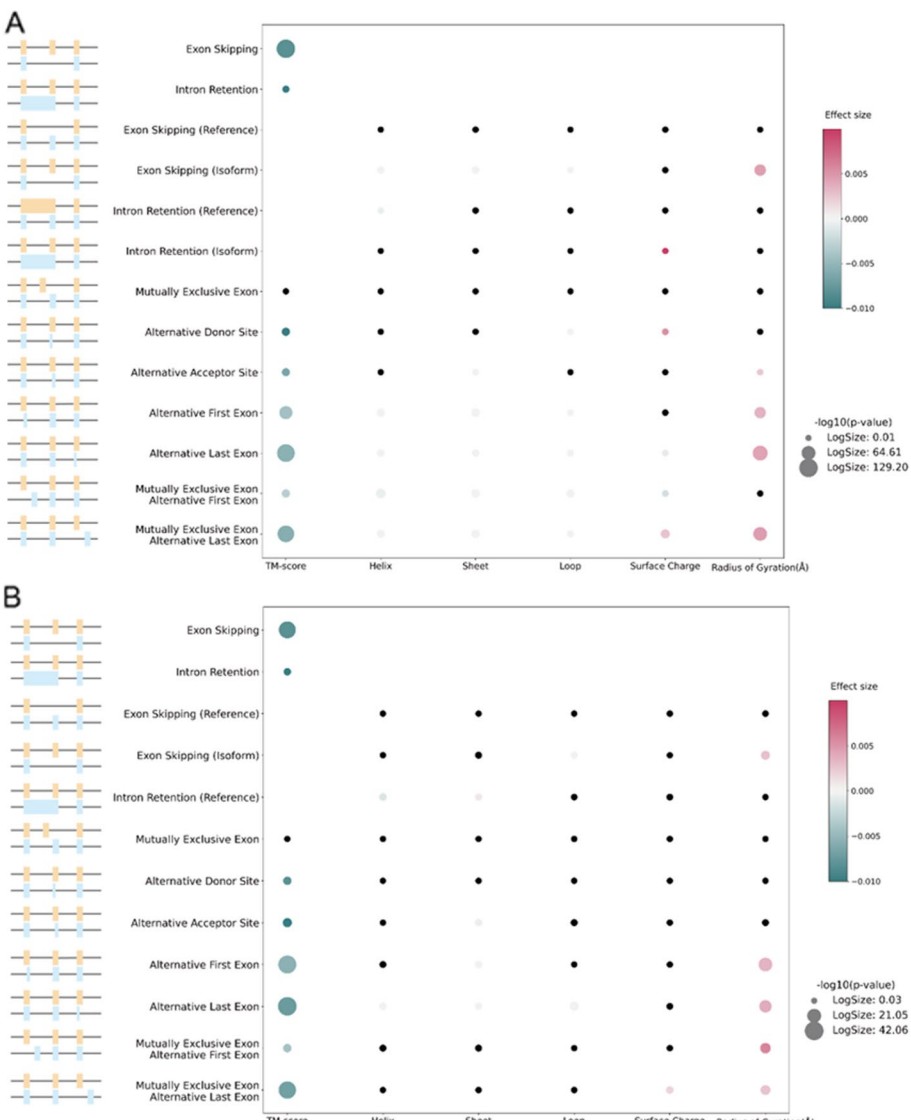

**Fig. 6** Per-residue effects of alternative splicing types on structural metrics. Per-residue effect analysis on our SwissProt dataset with high and confident prediction quality (**A**) and on the non-IDR, non-isolated helices dataset (**B**). Effect size is colored from green to pink to represent the negative to positive per-residue effect. The size of each dot represents the negative log10 *p*-value, and dots with insignificant *p*-value (*p*-value > 0.05) are colored black. The *p*-value for each effect size is obtained from the lm function in R

did not result in significant changes to secondary structure type percentage and surface charge. For the radius of gyration, ES and the four alternative splicing types occurring at the beginning or end of the protein showed positive per-residue effects toward radius of gyration (Fig. 6B).

In addition to the structural metrics mentioned above, we compared the percentage of IDRs between reference and alternate isoforms. For both structure and sequence-based methods of identifying IDRs, the distribution of IDR percentages are similar between references and alternate isoforms (Additional File 1:Fig. S9A) However, alternate isoforms exhibit a slightly higher median IDR percentage compared to references (reference median: 0.129, alternate isoform median: 0.147) based on window-averaged

pLDDT scores (*p*-value: 6.875e − 5, Mann–Whitney *U* test) (Additional File 1:Fig. S9B). The differences in IDR percentages indicate that reference and alternate isoforms of the same gene generally have comparable IDR content, with a median difference of 0 using window-averaged pLDDT scores and 0.001 using SPOT-Disorder (Additional File 1: Fig. S9C).

### Alternative splicing buries and exposes post translational modification sites

Protein function is regulated by the addition of various post translational modifications (PTMs) to amino acid residues, including phosphorylation, ubiquitination, acetylation, methylation, and glycosylation [48]. Modifications can lead to significant conformational changes, particularly in the case of phosphorylation. Furthermore, PTMs are added by enzymes that must be able to access the residue to be modified; for example, phosphorylation requires contact with specific kinases. Thus, phosphorylation sites buried inside the protein are more difficult for kinases to access compared to PTM sites that are solvent-accessible. If structure changes induced by alternative splicing modify the positions of PTM sites, these changes could have significant functional implications. Such changes are especially interesting in this study, because they cannot be directly predicted from sequence differences between reference and alternate isoform but can readily be predicted from structure. We therefore searched our predicted structures for evidence that alternative splicing can cause structural changes that bury or expose PTM sites.

To quantify changes in PTM site location, we calculated the relative solvent-accessible area (RSA) for each PTM site and determined whether the residue was solvent-accessible (exposed) or inaccessible (buried). We then grouped the PTM sites into five classes based on how they changed between reference and alternate isoforms: unchanged, spliced out, spliced in, buried to exposed, and exposed to buried. We identified a total of 587 exposed PTM sites in the reference structures that become buried in the alternate isoform and 1358 PTM sites that are exposed to the solvent in the reference but buried in the alternate isoform (Additional file 4:Table S3). For example, in the cell cycle protein Cyclin-dependent kinase 1 (CDK1), the 15-T phosphorylation site is buried in the reference (RSA: 20.8 Å$^2$), but is exposed (RSA: 175.6 Å$^2$) due to the loss of a helix in the alternate isoform (Fig. 7A). Similarly, the 200-T phosphorylation site is exposed (RSA: 79.1 Å$^2$) in the signaling pathway related protein TGF-beta receptor type-1 (TGFBR1) reference isoform, but the loop orientation buries (RSA: 16.5 Å$^2$) this site inside the TGFBR1 alternate isoform (Fig. 7B).

Such changes in the surrounding environment of PTM sites could be a mechanism by which alternative splicing regulates protein function. For example, the apoptosis regulator BAX (encoded by gene *BAX*) induces apoptosis under stress conditions, while phosphorylation at 184S by RAC-*α* serine/threonine-protein kinase (AKT1) causes BAX to prevent apoptosis [49]. In the BAX reference isoform (BAX-*α*), 184S is located inside a helix and the side chain is oriented toward the inside of the helix with the RSA of 1.18 Å$^2$. However, in the structure for the BAX-*δ* isoform, the corresponding residue 135S is exposed to the solvent with an RSA of 55.5 Å$^2$ (Fig. 7C). Moreover, the 184S site in BAX-*α* can form polar contacts with the residues from a parallel helix, while the corresponding residue (135S) in BAX-*δ* only has polar contacts with the residues in the same helix, which makes this site highly flexible in BAX-*δ* but rigid in

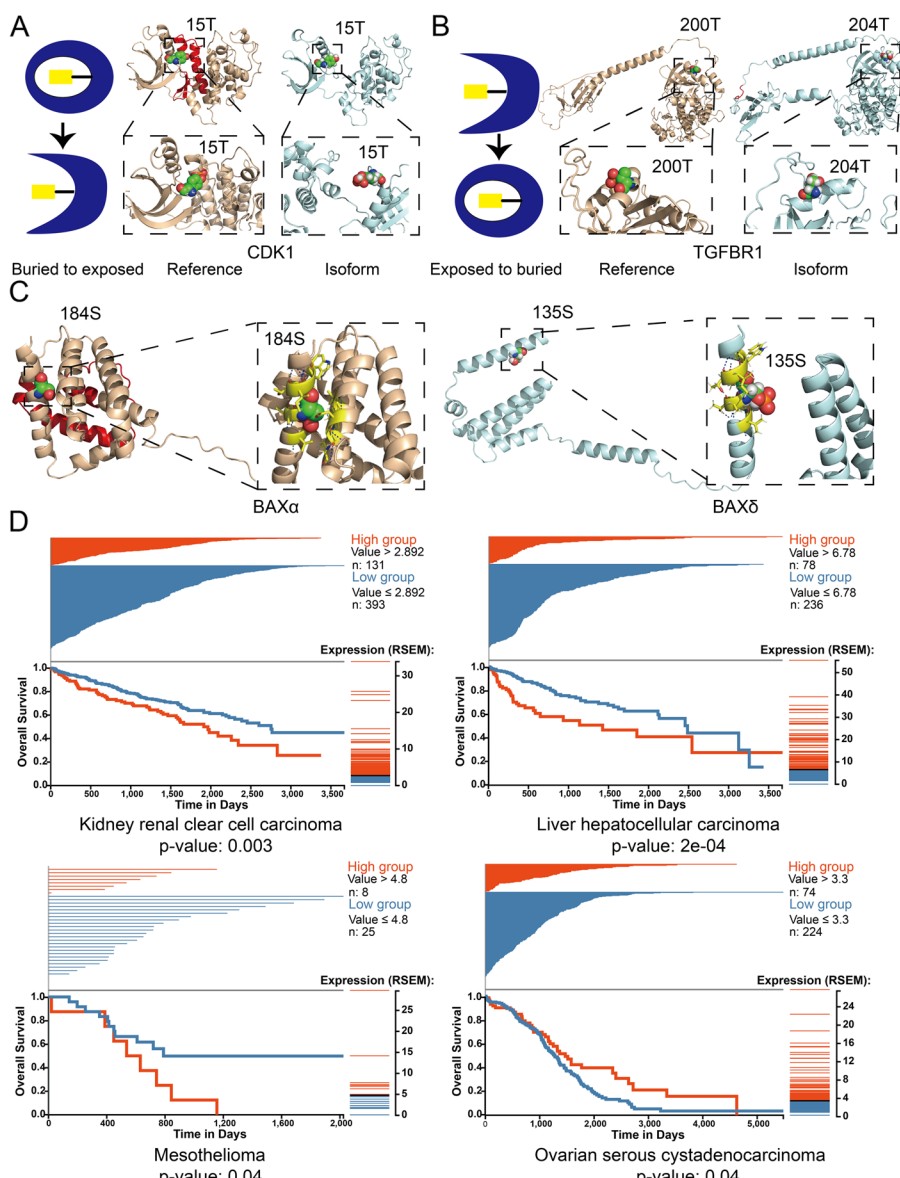

**Fig. 7** Alternative splicing buries and exposes post translational modification sites. **A** PTM sites changed from buried to exposed based on the threshold of relative solvent accessibility (RSA). Residue 15 T in the CDK1 reference isoform is buried (RSA: 20.8 Å²), and in the alternate CDK1 isoform, residue 15 T is exposed (RSA:175.6 Å²). **B** PTM sites changed from exposed to buried. Residue 200-T in the TGFBR1 reference isoform is exposed (RSA: 79.1 Å²), but buried. in the CDK1 isoform (RSA:16.5 Å²). Note that the extended single helix corresponds to a transmembrane domain. **C** The 184S residue is buried in BAX $\alpha$ (RSA: 1.18 Å²), and the corresponding PTM site 135S is exposed in BAX $\delta$ (RSA: 53.5 Å²). The residues within 4 Å of the PTM site are represented in yellow sticks, and the polar contacts with those residues are represented in blue lines. **D** Survival plot for patients with high vs. low expression of BAX $\delta$ in four cancer types. We split the sample into high and low expression groups using the 75th percentile and calculated the *p*-value using a log-rank test between high and low expression groups

BAX-$\alpha$ (Fig. 7C). In summary, the PTM site is almost completely inaccessible in BAX-$\alpha$ but highly accessible in BAX-$\delta$. Considering the critical role that phosphorylation of 184S plays in apoptosis regulation, this splicing-induced change in the structural context of the PTM site could have significant functional implications. Interestingly,

we identified several other differences in PTM site locations among BAX isoforms, including BAX $\zeta$ and BAX $\sigma$. (Additional File 1:Fig. S10). The structure of BAX $\zeta$ is similar to BAX-$\delta$: 106S, which corresponds to 184S in BAX-$\alpha$, has an RSA of 51.6 $\text{Å}^2$, and the polar contact is between the residues within the same helix (Additional File 1: Fig. S10A). While BAX $\sigma$ has a more compact structure, the PTM site (171S) is still exposed (RSA: 53.5 $\text{Å}^2$) and forms polar contacts within the helix (Additional File 1: Fig. S10B). Because of the protein's prominent role in regulating apoptosis, differences among BAX isoforms are especially interesting in the context of cancers. We investigated the expression of BAX isoforms across human cancer types using the TSVdb (TCGA Splicing Variants DB) [50]. We stratified donors based on high vs. low expression of the BAX-$\delta$ isoform across several cancer types (Fig. 7D). Compared with the low expression group, we observe significantly higher survival for kidney renal clear cell carcinoma, liver hepatocellular carcinoma, and mesothelioma but lower survival for ovarian serous cystadenocarcinoma in high expression BAX-$\delta$ isoform group (Fig. 7D).

The statistics for all metrics discussed so far including sequence identity, TM-score, secondary structure percentage, surface charge, radius of gyration, and PTM analysis are listed in Additional file 5:Table S4.

### Mapping isoform expression across human cell types

Using single-cell RNA sequencing (scRNA-seq) data, differential isoform usage has been discovered from a variety of cellular contexts, such as different mouse neuron types and also different human tissue compartments [21, 51]. An intriguing phenomenon lies in the dynamic switching of isoform usage within the same gene across diverse cell types, potentially shedding light on the distinctive functions of cell-type-specific protein isoforms. To investigate these effects, we conducted an analysis to pinpoint the predominant isoform usage within the same gene across 133 cell types, drawing from the Tabula Sapiens dataset. While many isoforms showed cell-type-specific expression, the majority of these involved differences in the expression level of the dominant isoform, rather than a true isoform switch in which an alternate isoform replaces the reference isoform as the highest-expressed form of a gene. Thus, we used stringent filtering criteria to identify genes showing true isoform switching. This revealed 33 genes, encompassing 71 modeled isoforms, exhibiting a notable shift in isoform usage across various cell types (Additional file 6:Table S5).

Qualitatively, these genes show clear cell-type-specific isoform switches (Fig. 8A). Some of the genes are well known examples of alternative splicing. For example, Myosin light polypeptide 6 (MYL6) expresses a non-muscle form and a smooth muscle form [52], and owing to its high expression level, it has been validated as differentially expressed between cell types from other scRNA-seq studies [51, 53, 54]. Analogous to MYL6, a gene associated with muscle function, tropomyosin (TPM), whose isoforms exhibit diverse binding dynamics with the actin filament [55], similarly demonstrates distinctive isoform usage. Specifically, we observe variations in isoform utilization concerning three members of the TPM family: TPM1, TPM2, and TPM3 (Additional file 6:Table S5).

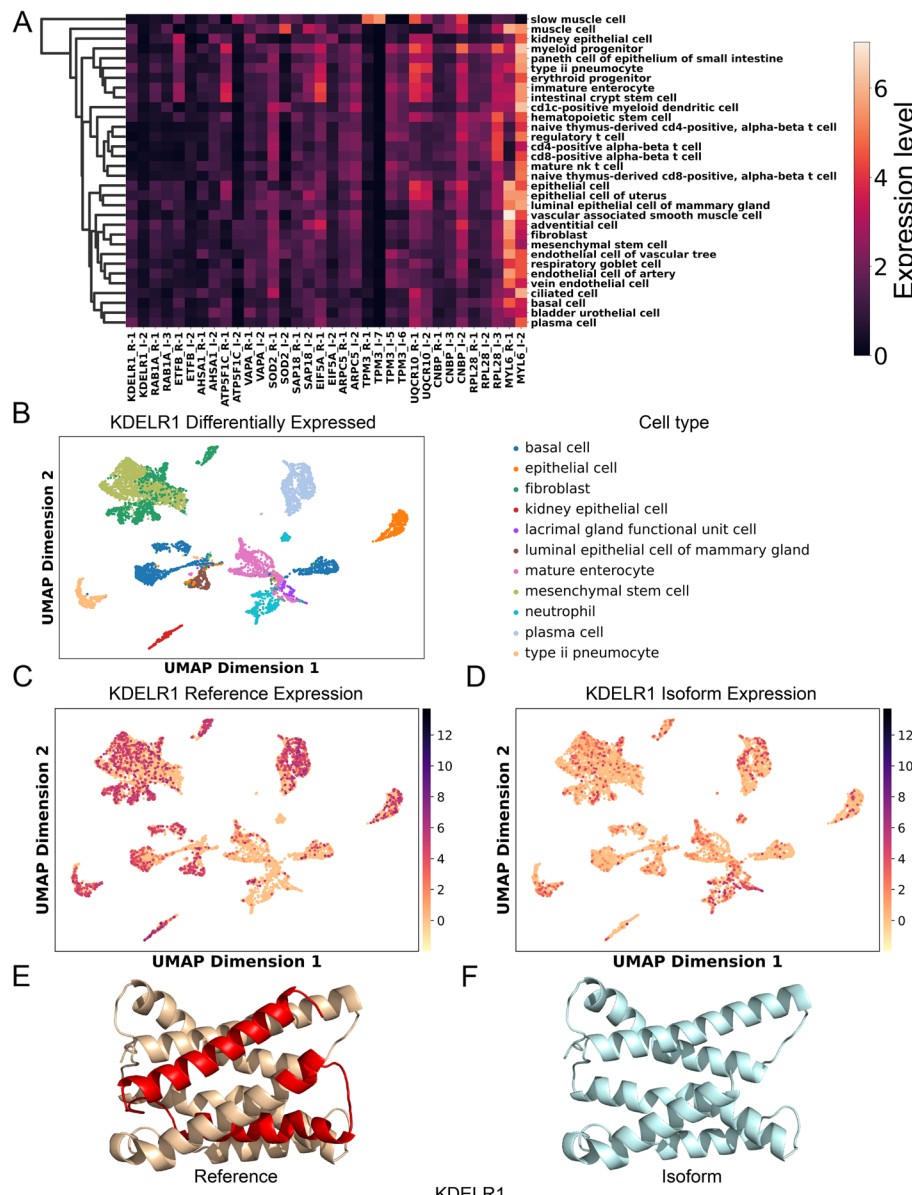

**Fig. 8** Cell-type-specific expression of alternate isoforms. **A** Expression profile for differential expressed reference/isoform across cell types. "R" in the label stands for each reference and "I" stands for an alternate isoform. **B** UMAP plot of the cell types with differentially expressed isoforms of gene KDELR1. **C, D** Expression of KDELR1 reference and isoform across eleven cell types. **E**, **F** Structure of KDELR1 reference and isoform. The alternatively spliced region is colored in red

Another example is the ER lumen protein-retaining receptor 1 (KDELR1), which functions as a receptor for KDEL signal-bearing proteins and facilitates the retrieval of proteins that have escaped from the endoplasmic reticulum (ER) [56]. In Fig. 8B, we identified cell types that differentially express the KDELR1 reference isoform and isoform 2. Compared to the KDELR1 reference isoform (Fig. 8C), isoform 2 shows high expression in specific cell types, including lacrimal gland functional unit cells and neutrophils (Fig. 8D). Although isoform 2 lacks 62 residues compared to the reference sequence, the overall structural fold is largely preserved (Fig. 8E), and importantly, the

lysine-rich motif critical for recognition is maintained [57, 58]. Another example is superoxide dismutase [Mn], mitochondrial (SOD2), where the SOD2 isoform 2 is highly expressed in cell types including muscle cell, stromal cell, and endothelial cell of vascular tree (Additional File 1: Fig. S11B). Interestingly, a critical residue (98H) manganese ion ($Mn^{2+}$) binding site is lost in the alternate SOD2 isoform, and it is believed to decrease the activity of isoform 2 [59]. We also observed that in the reference structure, the four $Mn^{2+}$ binding sites (50H, 98H, 183D, and 187H) are clustered together, while in the isoform structure, the remaining binding sites do not form a binding cluster, with 50H located away from the 144D and 148H (Additional File 1: Fig. S11C). Additionally, Ras-related protein Rab-1A (RAB1A) and phospholipid transfer protein (PLTP) also show significant differential expression between isoform and reference transcripts (Additional File 1: Fig. S11D-S11I). However, for the above isoforms, even though they have distinct expression patterns, without experimentally determined functional annotation, it is hard to link the differential expression, cell type function, and protein structure.

Interestingly, we also observed that genes which are differentially expressed among cell types like TPM3, OLFM3, and SOD2, also are differentially expressed in the same cell types across different tissues (Additional file 7:Table S6). We identified six genes with 12 isoforms that have distinct expression profiles in seven tissues. For example, among classical monocytes from different tissues, the TPM3 reference isoform has higher expression in blood, while TPM3 isoform 5 has higher expression in lung. Similarly, the SOD2 reference isoform has higher expression in endothelial cells from spleen, vasculature, trachea, and fat, while SOD2 isoform 2 is more highly expressed in skin endothelial cells.

### Structure-based function prediction nominates functional changes among spliced isoforms

We used COFACTOR [60] to predict functions for our isoform structures, allowing us to investigate the impact of alternative splicing on protein function. To do this, we quantified the number of gene ontology (GO) terms confidently predicted for each isoform, comparing these results with their respective reference isoforms. Function prediction shows that in general, reference and alternate isoforms have similar predicted functions (Additional File 1: Fig. S12A-S12B). In all three subcategories of gene ontology (biological process, cellular component, and molecular function), the reference isoforms exhibit a higher count of confidently predicted GO terms compared to the alternate isoforms, consistent with an overall trend toward loss of function in alternate isoforms. In order to determine if distinct alternative splicing types result in varying functional alterations of spliced isoforms, we conducted a hypergeometric test to assess the gain or loss of GO terms across all alternative splicing types. More GO terms are lost (935) in alternate isoforms compared to the gained (226) GO terms, and this difference is statistically significant ($p$-value < 0.05, hypergeometric test). Some GO terms are more likely than others to be gained or lost due to alternative splicing. For example, ADS is enriched in losing functions like response to chemical (GO:0042221), while AAS is likely to lose transcription regulator activity (GO:0140110) (Fig. 9A). Notably, AFE and MXE-ALE significantly lose terms related to the regulation of different processes. For gain of function, ADS is enriched for obtaining the organelle membrane term (GO:0031090), while ALE is enriched for gaining the intracellular membrane-bound organelle term (GO:0043231),

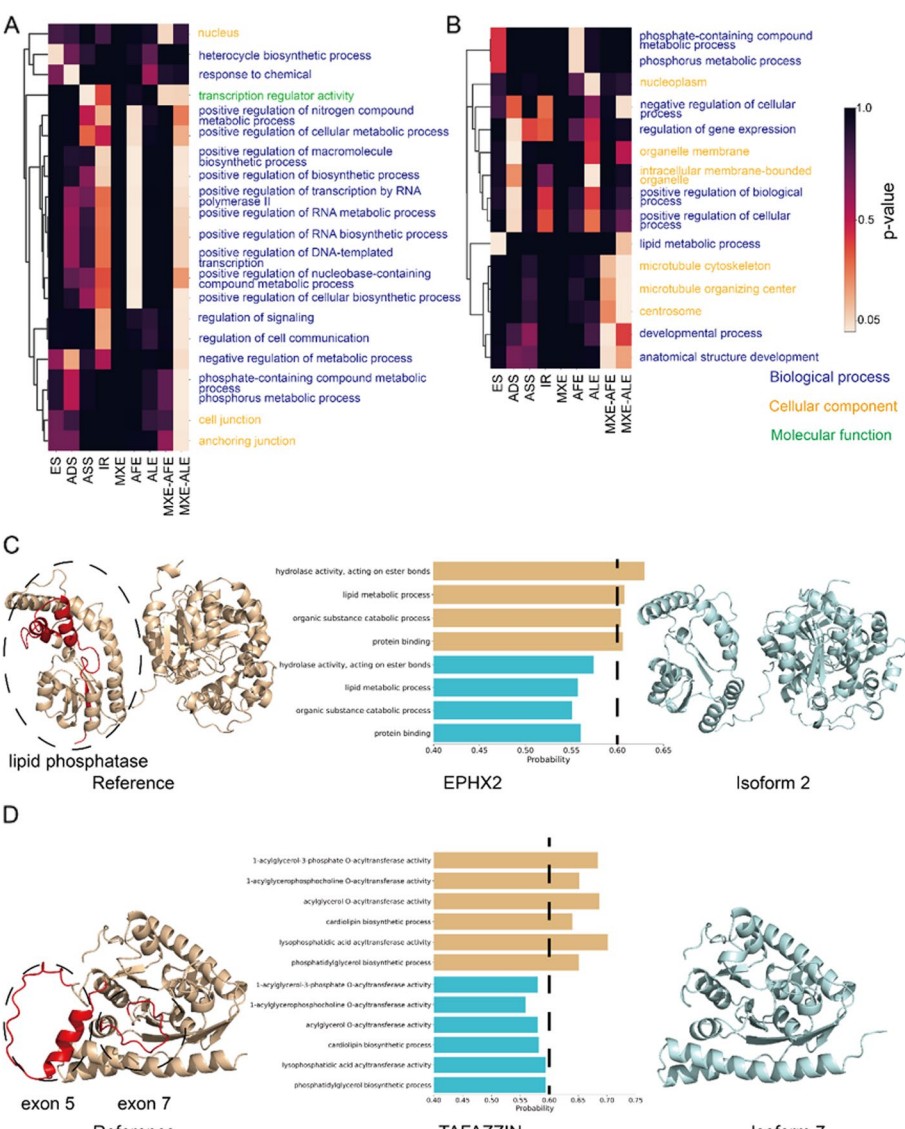

**Fig. 9** Structure-based prediction of functional changes induced by alternative splicing. Heatmap plot for GO terms frequently lost (**A**) or gained (**B**) across nine alternative splicing types, colored by *p*-value. The text colors for GO terms indicate which sub-ontology they come from. **C** Structures for EPHX2 reference, EPHX2 isoform 2, and predicted GO terms lost by EPHX2 isoform 2. The alternative splicing regions are colored in red. The lipid phosphatase domain is circled in the structure. **D** Structures for TAFAZZIN reference, TAFAZZIN isoform 7, and predicted GO terms lost by TAFAZZIN isoform 7. The alternative splicing regions are colored in red. The structures encoded by exon 5 and 7 are circled

which both are membrane-related GO terms (Fig. 9B). ES exhibits enrichment in gaining GO terms like lipid metabolic process (GO:0006629) and gaining of cytoskeleton-related terms including microtubule cytoskeleton (GO:0015630), microtubule organizing center (GO:0005815), and centrosome (GO:0005813). Alternative splicing at the beginning and end of the protein seems to have greater effect on predicted protein function, possibly because splicing at these positions leads to greater structural changes. This trend is consistent with the negative PCC between TM-score and sequence identity for alternative splicing types including AFE, AFE, ALE, MXE-AFE, and MXE-ALE in Fig. 3A. Also,

AFE, ALE, MXE-AFE, and MXE-ALE have longer sequence changes compared to other alternative splicing types (Additional file 8:Table S7).

We identified several interesting examples where isoforms gained or lost predicted functions compared to the reference isoform. Bifunctional epoxide hydrolase 2 (EPHX2) isoform 2 has a lower prediction probability for terms including hydrolase activity, acting on ester bonds (GO:0016788), lipid metabolic process (GO:0006629), organic substance catabolic process (GO:1901575), and protein binding (GO:0005515) (Fig. 9C). EPHX2 is a dual-function enzyme, with lipid phosphatase and epoxide hydrolase domains located at the N- and C-terminus, respectively [61]. Alternative splicing in isoform 2 removes the first 53 residues at the N-terminus. This structural change coincides with our functional predictions, suggesting that the lipid phosphatase activity may be affected by the loss of part of the N-terminal region. Another loss of function example is the TAFAZZIN isoform 7, which lacks exon 5 and exon 7 compared with the full-length reference, and is predicted to have lost transacylase activity [62]. The isoform is predicted to lose functions including 1-acylglycerol-3-phosphate O-acyltransferase activity (GO:0003841), acylglycerol O-acyltransferase activity (GO:0016411), lysophosphatidic acid acyltransferase activity (GO:0042171), and 1-acylglycerophosphocholine O-acyltransferase activity (GO:0047184) (Fig. 9D). Since the isoform that lost exon 5 is also functional, the loss of function of isoform 7 is likely due to the loss of exon 7, which seems to be important to the substrate-binding [63].

We also identified examples where the isoforms are predicted to gain functions compared to their references. RGN is a gluconolactonase, and we predict that RGN isoform 2 has more GO terms related to ester hydrolase activity (Additional File 1: Fig. S12C). The structure of the RGN reference is composed of five repeat domains, where isoform 2 lacks one domain, which may form a larger binding pocket and thus gain the ester hydrolase activity GO terms. Similarly, in another instance involving TGFB3, we predicted a gain in function, specifically in TGF II/III transforming growth factor beta receptor binding (GO:0007179 and GO:0034714), as a consequence of the loss of certain structural elements (Additional File 1: Fig. S12D). Notably, the region that is spliced out corresponds to the transforming growth factor beta-3 (TGF-beta-3) chains, while isoform 2 retains the structure of the latency-associated peptide and the signal peptide. This phenomenon underscores the concept that alternative splicing can, in certain instances, produce effects akin to the cleavage of a proprotein. Moreover, we also observed that some isoforms are predicted to gain some functions by inserting some unstructured regions. In examples like TRAF2 isoform 2 (Additional File 1: Fig. S12E) and PCID4 isoform 4 (Additional File 1: Fig. S12F), the gained GO terms are related to cellular response and regulation. The inserted flexible loop may be a key factor of the gained GO terms, since some intrinsic disorder regions can mediate new protein–protein interactions (PPIs) [64].

## Discussion

Alternative splicing is recognized as a key driver of protein diversity [1, 65], but there is an ongoing debate concerning the functional significance of these spliced isoforms [66]. When splicing occurs at domain boundaries or simply destabilizes a fold, this causes a change in abundance of the functional unit of the protein, similar to

regulation of gene expression by transcription factors, chromatin exposure, etc. However, unlike regulation of gene expression, alternative splicing may change the structure to modulate the protein function in more complex ways, including modifying the molecular function, cellular localization, and/or regulation through interactions with other biomolecules [67, 68]. Guided by the "sequence-structure–function" paradigm, we folded human spliced isoforms and applied various structural and functional metrics to reveal the functional implications of alternative splicing from a structural view. Our discoveries contribute to an enhanced comprehension of the sequence determining protein structures. However, it is noteworthy that we observed a generally lower evolutionary conservation suggested by MSAs in splicing regions, which may have a potential impact on structure prediction. This concern is not unique to alternative splicing regions; it pertains to all protein regions with insufficient homologous sequences. It is valuable to explore MSA-enhanced methods and leverage large protein language models to enhance the prediction quality for sequences with limited MSAs, including some alternative splicing regions [69].

For large-scale data analysis, the identification of outlier instances proves to be a more valuable approach than the generation of broad, statistically significant conclusions. This preference arises due to the susceptibility of the latter to bias resulting from the abundance of data points exhibiting subtle distinctions between the two categories. Notably, we have successfully pinpointed specific isoforms within our dataset whose structural features underwent substantial alterations due to alternative splicing, as evidenced in Additional file 3:Table S2. It is worth highlighting that the functional characteristics of the majority of these outlier isoforms have yet to be characterized, thereby implying that our methods may have preemptively selected certain spliced isoforms with the potential for functional divergence from the reference form.

While AlphaFold2 has been demonstrated to be powerful and broadly accurate, a limitation of this computational analysis is that predictions may be biased and inaccurate. Specifically, MSAs for alternatively spliced regions may be shallower, leading to lower quality predictions for alternatively spliced regions; since AlphaFold2 is trained on experimentally characterized structures often at cryogenic temperatures, the predictions may be biased toward folded structures; we only use a single AlphaFold2 predicted conformation, which may not be representative of the dynamic protein function; and we are only predicting protein monomers while many proteins function in complex with other proteins and biomolecules. While we make efforts to investigate and control for these factors, including assessing the difference in predicted confidence across sequence length, and differences in predicted intrinsic disordered regions, and look at numerous specific examples, further analysis and experimentation will refine these calculations. These limitations, however, should not detract from the overall contributions of the work—namely that global prediction of isoforms enabled by this study can systematically give information about the structural and functional impact of alternative splicing, an important dimension of the sequence to function relationship.

Combined with the isoform-level scRNA-seq expression data, our findings could help to enhance the annotation of the selected canonical/reference sequences from UniProt/SwissProt database. Additionally, our approach allows for a more

fine-grained elucidation of expression data at both cell-type and isoform levels, despite the disparities between mRNA and protein expression levels [70]. Furthermore, we have established connections between the predicted structural features and their expression profiles within various cell types and tissues, with the overarching objective of shedding light on the distinctive cellular perspectives offered by protein structural analysis.

In light of recent advancements in computational tools, exemplified by AlphaFold2, this research endeavor stands as a pioneering model for harnessing AlphaFold2's capabilities in exploring pertinent structural themes. To enrich our comprehension of alternative splicing, future investigations may pivot toward the exploration of protein–protein interactions (PPI) [71, 72], domain analysis and isoform function prediction [73–75], aimed at facilitating the annotation and elucidation of functional spliced isoforms.

## Conclusions

In this study, we predicted the structures of over 11,000 human alternative splicing isoforms using AlphaFold2 and applied structural metrics to compare reference and spliced isoforms. Most structural differences were driven by sequence variation, though some isoforms showed major structural changes despite high sequence similarity. Using the BAX isoform as an example, we demonstrated how alternative splicing can alter the solvent environment of post-translational modification sites and influence isoform-specific functions. Finally, integrating predicted structures with single-cell expression data allowed us to identify how structurally varying isoforms are differentially expressed across cell types. Overall, our work provides a systematic structural perspective on the functional impact of human alternative splicing.

## Methods

### Structure prediction of human isoforms

We gather the alternative splicing sequences from the 2021_09 release of UniProt database [76], with a particular emphasis on SwissProt, where the data are manually curated and information for spliced isoforms is available. To enhance computational efficiency and mitigate memory constraints, we restrict our selection to sequences containing fewer than 600 amino acids. For each gene, we selected the SwissProt canonical sequence as the reference sequence, considering any supplementary sequences listed in the "Sequence & Isoforms" section as isoform sequences. We ran AlphaFold2 locally (version: 2.2.0), utilizing default parameter settings, to generate predictions for spliced isoforms [16]. We obtain reference structures for each protein from the AlphaFold Protein Structure Database, 2022_01 release (https://alphafold.ebi.ac.uk/) [27] as provided by EBI. For both the reference and isoform structures, we computed multiple sequence alignment (MSA) locally, by extracting them from the feature pickle file before the prediction stage of AlphaFold2. The effective MSA depth is calculated for each isoform and reference sequence based on formula (1) [77]:

$$Neff = \sum_{i=1}^{N} \frac{1}{1 + \sum_{j=1, j \neq i}^{N} 1(I_{i,j} \geq 0.8)} \tag{1}$$

where *Neff* is the number of effective MSA for each query protein sequence, $N$ is the total number of MSA for the query sequence, $1(I_{i,j} \geq 0.8)$ means the sequence identity will be calculated for each two sequences, and if they have sequence identity higher than 0.8, this value will be 1, otherwise will be 0.

We collected a total of 5966 reference structures from EBI and predicted 11,159 isoform structures. Structures are divided into four datasets based on their mean per-residue confidence score (pLDDT). Only structures with high (pLDDT > 90) and confident (70 < pLDDT < 90) quality are utilized for metrics analysis, which results in 4450 reference structures and 7631 isoform structures.

For comparison, we also obtained the CHESS (version: 1.1) human protein structure database from (https://www.isoform.io/) [19]. In this database, Sommer et al. employed AlphaFold2 to predict isoform structures at a transcript level, with a maximum sequence length of 1000 amino acids. To determine the reference sequences for the CHESS dataset, we use both the Matched Annotation from NCBI and EMBL-EBI (MANE) GRCh38 v0.95 file and the CHESS v2.2 gene annotation file [28, 78]. In total, we gathered 15,727 reference structures and 97,824 isoform structures. Among these, 7923 isoforms from our SwissProt dataset could be matched to the CHESS dataset, and we excluded 81,229 structures for which we could not decide whether they were isoforms or references. Similarly, for metrics analysis, only 10,597 reference structures and 57,973 isoform structures with high or confident prediction quality were included.

To validate AlphaFold2's prediction ability on isoform sequences, we employed local Protein BLAST (version: 2.12.0 +) to search against all Protein Data Bank (PDB) database sequences. We selected experimental isoform structures with an E-value threshold of < 10e − 5. We manually excluded the structures which (1) included unmodeled splicing regions; (2) were from non-human species; and (3) were low resolution structures from NMR and CryoEM. To assess structural similarity, we used TM-score and RMSD based on US-align (version: 20,230,609), which included the TM-align to compare AlphaFold2-predicted structures with experimental structures [79], and we used the "-seq" option to ensure the sequence aligned correctly before the structure alignment. Furthermore, we conducted a paired *t*-test to compare reference and isoform structures in terms of TM-score and RMSD. Notably, the AlphaFold2 structures used for comparison were predicted in a template-free setting of AlphaFold2.

### Algorithm for identifying alternative splicing types

To map the exons between each isoform and its respective reference sequence, we use the human genome release 40 (GRCh38.p13) (Ensembl 106) General Transfer Format (.gtf) file provided by Gencode. We match the SwissProt accession numbers with their corresponding Ensembl transcript numbers using the SwissProt metadata file from Gencode. It should be noted that 3252 SwissProt entries (reference or spliced isoform) lack Ensembl accession numbers, and also 484 entries are labeled as nonsense-mediated decay. Consequently, we exclude these isoforms from our analysis of alternative splicing types. We successfully determine the alternative splicing types for 7685 isoforms from our SwissProt isoform structure dataset and 97,824 isoforms from the CHESS dataset. The exon position information is matched and compared pairwise between each isoform transcript and corresponding reference transcript. Different from the seven alternative

splicing events defined in previous study [8], we designed our methods to detect nine alternative splicing events: exon skipping (ES), alternative donor site (ADS) or alternative 5′ splice site (A5S), alternative acceptor site (AAS) or alternative 3' splice site (A3S), mutually exclusive exons (MXE), intron retention (IR), alternative first exon (AFE), alternative last exon (ALE), mutually exclusive exon-alternative first exon (MXE-AFE), and mutually exclusive exon-alternative last exon (MXE-ALE) [80, 81]. Given that a prior study observed that alternative first exons can display mutually exclusive behavior [82], we introduced the definitions of mutually exclusive exon-alternative first exon (MXE-AFE) and mutually exclusive exon-alternative last exon (MXE-ALE) to differentiate them from alternative first exon (AFE) and alternative last exon (ALE). The graphical representations of each alternative splicing type, along with illustrative example structures, are depicted in Additional File 1: Fig. S1.

### Metrics for structural analysis

We compute the TM-score using the tmtools Python package (version: 0.0.2) based on TM-align [83]. In each iteration, we perform two alignments of isoform-reference pairs, utilizing either the reference or isoform as the template. The average of the two TM-scores is considered as the final TM-score, accounting for potential disparities between SwissProt selected reference sequence and the predominant functional isoform within the human body.

We assess sequence identity between each isoform sequence and its corresponding reference sequence using pairwise global sequence alignment from the Biopython (version: 1.79) package [84]. Initial global alignments employ the BLOSUM62 substitution matrix and the Needleman-Wunsch algorithm. Subsequently, sequence identity is computed based on the obtained alignments, see formula (2).

$$I_s = N_i/L_a \tag{2}$$

where $I_s$ is the sequence identity between reference and isoform sequences, $N_i$ is the number of identical residues, $L_a$ is the alignment length.

We measure the Pearson correlation coefficients (PCC) between the TM-score and sequence identity for each isoform-reference pair. The residue contact maps shown in Fig. 3C,D are computed using the ProteinTools [85].

We use PyMOL (https://pymol.org/2/) (version: 2.4.1) to calculate DSSP secondary structure percentage using the dss command [86]. In our analysis, we only consider the three primary secondary structure types: helix, sheet, and loop.

For surface charge comparison, we first apply the PDB2PQR30 program (version: 3.5.2) with the PARSE force field to assign charges to each residue [87]. Subsequently, to get the surface residues, we calculate the solvent accessible surface area (SASA) using the "measure sasa" command within Visual Molecular Design (VMD) (https://www.ks.uiuc.edu/Research/vmd/) (version: 1.9.2), with the restrict distance set as 1.4 Å. We then

use the relative solvent accessibility (RSA) to identify surface residues, which is usually used to decide the exposure content of a residue [88]. RSA is SASA after normalized by the maximum allowed SASA value, and we apply a threshold RSA as defined by Tien et al. [88], where residues with an RSA higher than the threshold are considered surface residues. The surface charge for each protein is determined by summing the charges of all surface residues.

We calculate the radius of gyration for each isoform and reference structure using the VMD according to the formula (3):

$$R(g) = \sum_i m_i \cdot (r_i - r_c)^2 / \sum_i m_i \tag{3}$$

where $m_i$ represents the mass for atom i, $r_i$ represents the radius for atom i, and $r_c$ represents the radius of the center of mass.

We use the lm function in R (version: 4.1.1) to construct a linear regression model to study the effect of specific alternative splicing types. Our features are derived from the sequence length results for each alternative splicing type within each isoform. For alternative splicing types like ES and IR, which could happen in both isoform and reference, we separate them into ES in reference, ES in isoform, IR in reference, and IR isoform to distinguish the alternative splicing in reference and isoform. For secondary structure percentage, surface charge, and radius of gyration, we use the difference between the isoform and reference as the outcome variable to build the linear regression model (formula (5)). In the case of TM-score, which inherently involves a comparison between two structures, we employ the absolute length difference caused by each alternative splicing event as the feature, and since it is non-directional, we do not specify the situation where ES and IR occur in reference and isoform (formula (7)). All the predicted metrics are standardized to mean 0 and standard deviation of 1. The coefficient associated with each alternative splicing type was interpreted as the per-residue effect, and the $p$-value for each coefficient is obtained from the lm function.

$$\Delta L_{AS} = \Delta L_{ADS} + \Delta L_{AAS} + \Delta L_{MXE} + \Delta L_{AFE} + \Delta L_{ALE} + \Delta L_{MXE-AFE} + \Delta L_{MXE-ALE} \tag{4}$$

$\Delta L_{AS}$ represents the sequence length change (Isoform-reference) from the following alternative splicing types: ADS, AAS, MXE, AFE, ALE, MXE-AFE, MXE-ALE.

$$\Delta_M = \Delta_{pLDDT} + \Delta L_{ES(R)} + \Delta L_{ES(I)} + \Delta L_{IR(R)} + \Delta L_{IR(L)} + \Delta L_{AS} \tag{5}$$

$\Delta_{pLDDT}$ is the difference of pLDDT. $\Delta_M$ represents the difference of metrics including secondary structure percentage, surface charge, and radius of gyration. $\Delta L_{ES(R)}$, $\Delta L_{ES(I)}$, $\Delta L_{IR(R)}$, and $\Delta L_{IR(I)}$ are the sequence length change caused by ES in reference, ES in isoform, and IR in reference and IR in isoform.

$$|\Delta L_{AS*}| = |\Delta L_{ADS}| + |\Delta L_{AAS}| + |\Delta L_{MXE}| + |\Delta L_{AFE}| + |\Delta L_{ALE}| + |\Delta L_{MXE-AFE}| + |\Delta L_{MXE-ALE}| \tag{6}$$

$|\Delta L_{AS*}|$ represents the absolute value of the sequence length change from the following alternative splicing types: ADS, AAS, MXE, AFE, ALE, MXE-AFE, MXE-ALE.

$|\Delta L_{ES}|$, $|\Delta L_{IR}|$ are the absolute value of sequence length caused by ES and IR, respectively.

$$TM - score = |\Delta_{pLDDT}| + |\Delta L_{ES}| + |\Delta L_{IR}| + |\Delta L_{AS*}| \tag{7}$$

### Intrinsically disordered region analysis

We identify IDRs for both reference proteins and their isoforms using two approaches: window-averaged pLDDT scores of AlphaFold2 predicted structures [29] following the criteria described in Tesei et al. [35] and SPOT-Disorder, a sequence-based prediction method [89]. For the window-averaged pLDDT score approach, we compute IDRs for all predicted structures. Since SPOT-Disorder takes 1–3 h per structure, we only compute IDR predictions for the SwissProt dataset with high and confident prediction quality. For each structure and each method, we calculated the IDR percentage as the ratio of IDR length to the total protein length and summarized results in Additional file 5:Table S4. The IDR percentage difference shown in Additional File 1: Fig S9C is calculated as the IDR percentage of each isoform minus the IDR percentage of its corresponding reference.

We also create a non-IDR dataset using only window-averaged pLDDT scores by selecting isoforms and their references that have no IDRs. This resulted in 1773 non-IDR isoforms, 1770 of which overlap with the high-confidence dataset. Since disordered regions have variable radius of gyration, we restrict the radius of gyration analysis to this non-IDR dataset.

### Identifying structures with isolated helices

To identify isolated helical structures, a known artifact of AlphaFold2 predictions, we first search for regions where residues are both helical and solvent-exposed, based on relative solvent accessibility (RSA). To be consistent with the IDR analysis, we define isolated helices as continuous regions of more than 20 exposed helical residues. This analysis is performed only on the non-IDR dataset. After filtering, we obtain a dataset that excludes both IDRs and isolated helices, comprising 1359 isoforms and 975 reference proteins. All structural analyses are conducted on this filtered dataset, and the results are shown in Additional File 1: Fig. S4 and Fig. 6B.

### Analysis of PTM sites

We collect post translational modification (PTM) site information from Phospho-SitePlus (release 2022.07) [90], which includes seven main PTM types: phosphorylation, ubiquitination, acetylation, methylation, O-GalNAc glycosylation, O-GlcNAcylation, and sumoylation. To accurately classify PTM sites, we first map the alternative splicing regions for each isoform based on the information from SwissProt; PTM sites located outside these regions are classified as "retained." Similarly, PTM sites in the regions labeled as "Missing" are categorized as "spliced out" PTM sites. And also, there are PTM sites from the PhosphoSitePlus that have records only in isoforms. We reverse map these sites back to their reference counterparts, classifying sites that could not be matched to

the reference as "spliced in" PTM sites. For the "retained" PTM sites, we use the RSA to determine the exposure extent for every PTM site, classifying them as either "buried" or "exposed." If the PTM site is consistently buried or exposed between reference and isoform, it will be labeled as "unchanged." In total, we classify the PTM sites into five categories: unchanged, spliced out, spliced in, buried to exposed, and exposed to buried. It is worth noting that due to the absence of protein functional annotation for the CHESS isoform structure dataset, PTM site analysis is exclusively applied to our SwissProt isoform structure dataset.

We obtain the RNA-seq data for BAXα and BAXδ from the TCGA Splicing Variants DB, which measures expression across 33 cancer types [50]. For both BAXα and BAXδ, we split them into high and low expression groups based on their expression in each cancer type by a ratio of 0.25:0.75 (high:low). We use a log-rank test with a threshold *p*-value of 0.05 between high and low expression groups to determine whether expression of specific BAX isoforms will affect survival or not.

Visualization of the phosphorylation form of PTM sites is simulated by the SwissSidechain (version: 2) [91].

### Preprocessing single-cell data

To reliably identify alternative splicing from single-cell RNA sequencing (scRNA-seq) data, we gather the fastq files for the Smart-seq2 dataset from Tabula Sapiens Consortium (https://tabula-sapiens-portal.ds.czbiohub.org/) [23]. Then, we use the Kb-python (version: 0.27.3) to generate the transcript count matrix from fastq files [92]. We build the index using the "kb −ref" command from kb-python, using the Human genome release 39 (GRCh38.p13) gtf file along with the CHESS2.2 gtf file as the input annotation files. We use the "kb -count" command from kb-python to generate the count matrix. For cell type annotations, we obtain the information for each cell from https://figshare.com/projects/Tabula_Sapiens/100973. Cells without cell type annotations are excluded, resulting in a dataset comprising a total of 133 cell types. We follow the normalization procedure outlined by Sina et al. [21], which involves normalizing the expression of each transcript by its length and the library size, calculating the transcripts per million (TPM) for each transcript, and applying a log transformation to the expression data. After preprocessing, we generate two cell-by-transcript matrices for both the Gencode and CHESS transcripts, consisting of 26,748 cells and 164,607 transcripts for Gencode and 26,748 cells and 113,551 transcripts for CHESS, respectively. The cell-by-gene matrix is constructed by aggregating the expression of all transcripts within each gene, resulting in 26,748 cells and 19,980 genes for the Gencode dataset and 26,748 cells and 18,659 genes for the CHESS dataset.

### Isoform usage shift analysis of single-cell data

We specifically subset a SwissProt count matrix from the Gencode dataset (after normalization), including only the isoforms for which we have modeled the protein

structures. This subset consists of 26,748 cells and 13,524 isoforms and is used for the analysis of differentially expressed isoforms. For all isoforms within each gene, we conduct either a *t*-test (for two isoforms) or a one-way analysis of variance (ANOVA) test (for more than two isoforms). The purpose is to assess whether there are significant differences in the expression levels for each isoform within the same gene. We then identify transcripts that exhibit expression specific to particular cell types below the critical *p*-value threshold of 0.05, after applying the Bonferroni correction. Different isoforms of the same gene which are upregulated for the same cell type are excluded, and we also exclude the cell types with fewer than 30 cells; the results are sorted based on fold change. Additionally, we perform the same differential expression analysis on 45 heterogeneous cell types originating from various tissues. This analysis is aiming to identify isoforms that could distinguish expression patterns within the same cell type across different tissues.

### Structure-based function prediction with COFACTOR

The GO terms for the human proteins are predicted by a modified version of COFACTOR [93] optimized for large-scale structure-based function annotation. The pipeline consists of four complementary pipelines based on sequence, structure, protein–protein interactions (PPIs), and Pfam domain family.

In the sequence-based pipeline, the query sequence is searched by BLASTp through the UniProt Gene Ontology Annotation (UniProt-GOA) database with default parameters to identify templates with GO annotations. Only annotations validated by experimental or high-throughput evidence, traceable author statement (evidence code TAS), or inferred by curator (IC) are considered. The prediction score of GO term *q* is defined as:

$$Cscore_{sequence}(q) = \frac{\sum_{k-1}^{K(q)} ID_k(q) \cdot bitscore_k(q)}{\sum_{k-1}^{K} ID_k \cdot bitscore_k} \tag{8}$$

Here, *K* is the total number of BLASTp hits; $bitscore_k$ is the bit-score of the *k*th hit; $ID_k$ is the global sequence identity of the *k*th hit; $K(q)$ and $bitscore_k(q)$ are the corresponding values for the subset of BLASTp hits with GO term *q*.

In the structure-based pipeline, the query sequence is searched by Foldseek [94] (release 2023_06) using parameters "–tmalign-fast 1 -e 10" through the subset of the AlphaFold Structure Database with GO annotations in UniProt-GOA. The identified hits are re-aligned by TM-align [83] to get the TM-score. The prediction score is as:

$$Cscore_{structure}(q) = \frac{\sum_{k-1}^{K(q)} ID_k(q) \cdot TM_k(q) \cdot bitscore_k(q)}{\sum_{k-1}^{K} ID_k \cdot TM_k \cdot bitscore_k} \tag{9}$$

Here, $TM_k$ is the TM-score obtained from TM-align realignment of the *k*th Foldseek hit; $TM_k(q)$ is the TM-score for Foldseek hit with term *q*.

In the PPI-based pipeline, the query protein is mapped to the STRING [95] PPI database to identify PPI partners with GO annotation. The prediction score can be calculated as:

$$Cscore_{ppi}(q) = \frac{\sum_{k-1}^{K(q)} string_k(q)}{\sum_{k-1}^{K} string_k} \tag{10}$$

Here, $string_k$ is the STRING score of the $n$th PPI partner, and $string_k(q)$ is the STRING score for the partner with term $q$.

In the Pfam-based pipeline, the query protein is searched by hmmsearch through the Pfam database to get the list of Pfam hits. Only the 7000 most common Pfam families in the UniProt database are considered. The prediction score for term $q$ is derived by logistic regression.

$$Cscore_{pfam}(q) = \frac{1}{1 + exp(-w_0 - \sum_{m=1}^{7000} w_m \cdot x_m)} \tag{11}$$

Here, $x_m$ indicates the match of the $m$th Pfam family; $w_m$ is the weight optimized on a separate training set.

The prediction scores of the sequence-, structure-, PPI-, and Pfam-based pipelines are combined with the prediction score from a recent deep learning-based GO predictor and the background frequency of the GO term in the UniProt-GOA as six different input features [96]. These features are fed into a gradient boosted tree model trained by LightGBM to obtain the final consensus GO prediction [97].

We used as probability of 0.6 as the threshold; only GO terms with above 0.6 predicted probability were considered as confident GO terms. The alternative splicing type-wise GO enrichment hypergeometric test was calculated by the following formula using the scipy python package (version: 1.7.1) on our high and confident dataset:

$$p(k, M, n, N) = (\frac{\frac{n!}{k! \cdot (n-k)!} \cdot \frac{(M-n)!}{(N-k)! \cdot ((M-n)-(N-k))!}}{\frac{M!}{N! \cdot (M-N)!}}) \tag{12}$$

where $k$ represents the number of isoforms with gain/lose this GO term and with this AS type, $M$ represents the total number of isoforms, $n$ represents the number of isoforms which gain/lose this GO term, and $N$ represents the number of isoforms with this AS type.

For heatmaps shown in the figures, we removed the go terms with too few gained or lost terms (minimum of 5 gained GO terms and minimum of 15 lost GO terms), and we also removed general GO terms which have over 5000 predictions among isoforms and references with high and confident prediction quality.

The UMAP embedding is calculated using the umap-learn (version: 0.5.3) scipy package based on a pre-computed Jaccard distance matrix for the confident GO terms each reference/isoform contained.

## Supplementary Information

---

Additional file 1: Supplementary Fig. S1-S12.

Additional file 2: Table S1. Comparison of AlphaFold2-predicted and X-ray resolved structures for alternatively spliced isoforms.

---

Additional file 3: Table S2. Examples of alternatively spliced isoforms with high sequence identity but low TM-score.

Additional file 4: Table S3. Summary of post-translational modification (PTM) site changes between reference and isoform proteins.

Additional file 5: Table S4. Summary of structural metrics from pairwise comparisons between reference and isoform proteins.

Additional file 6: Table S5. Summary of isoforms usage shifts across cell types.

Additional file 7: Table S6. Differential isoform usage across tissues for the same cell type.

Additional file 8: Table S7. Average sequence length changes across nine alternative splicing types.

Additional file 9: Peer review history.

### Acknowledgements
We thank members of the Freddolino lab and Matthew Karikomi for helpful discussions and feedback.

### Review history
The review history is available as Additional file 9.

### Declaration of generative AI and AI-assisted technologies in the writing process
During the preparation of this work, the authors used ChatGPT in order to polish style and grammar. After using this tool, the authors reviewed and edited content as needed. We take full responsibility for the content of this publication.

### Peer review information

### Authors' contributions
YS: Methodology, Software, Formal analysis, Investigation, Data Curation, Writing—Original Draft, Visualization. CZ: Methodology, Investigation, Formal analysis, Writing—Review & Editing. GSO: Discussion, Review & Editing. MJO: Conceptualization, Methodology, Formal analysis, Writing—Review & Editing, Supervision, Project administration. JW: Conceptualization, Methodology, Formal analysis, Writing—Review & Editing, Supervision, Project administration, Funding acquisition.

### Funding
This work was supported by Chan-Zuckerberg Initiative grant PN-0000000075 to J.D.W. G.S.O. acknowledges support from National Institutes of Health Grants P30 ES017885-11-S1 and U24 CA271037, and M.J.O acknowledges support from National Institutes of Health grant R35GM151129.

### Data availability
The alternative splicing sequences used for structure prediction are gathered from the 2021_09 release of the UniProt database (https://www.uniprot.org/) [76]. The predicted alternative isoform structures are predicted by AlphaFold2 (version: 2.2.0) [16]. The reference isoform structures are gathered from the 2022_01 release of AlphaFold Protein Structure Database (https://alphafold.ebi.ac.uk/) [16, 27]. The structures from the CHESS dataset are downloaded from (https://www.isoform.io/) (version: 1.1) [19]. The post translational modification (PTM) site information is gathered from PhosphoSitePlus (https://www.phosphosite.org/) (release 2022_07) [90]. The Smart-seq2 scRNA-seq is from Tabula Sapiens v1 (https://tabula-sapiens-portal.ds.czbiohub.org/) [23]. All scripts and analysis code are available on GitHub (https://github.com/welch-lab/AF2_scRNA) and Zenodo (https://doi.org/10.5281/zenodo.16660869) [98], with MIT license. All predicted structures are available via Figshare: https://doi.org/10.6084/m9.figshare.24891870, processed human scRNA-seq data is available via Figshare: https://doi.org/10.6084/m9.figshare.24843948, function prediction results are available via Figshare: https://doi.org/10.6084/m9.figshare.24891897.

## Declarations

### Competing interests
The authors declare no competing interests.

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

## 