## [Additional file 9: Peer review history. · Genome Biology]

Review history

First round of review

Reviewer 1

In the manuscript "Predicting the Structural Impact of Human Alternative Splicing" by Song et al., the authors present a systematic analysis of AlphaFold models of protein isoforms generated through alternative splicing. This is a significant area of research that has become feasible on a large scale due to advancements provided by the AlphaFold program.

The structural models predicted by AlphaFold form the foundation of this research; therefore, their quality and robustness must be assured. The authors should make additional efforts to convince readers of the models' accuracy. Specifically, I highlight two issues for consideration.

First, in several figures throughout the manuscript, the structures of certain isoforms contain alpha-helices that lack extensive interactions with the rest of the structure. Such occurrences are extremely rare in experimentally determined 3D structures (almost non-existent), yet AlphaFold predicts these isolated alpha-helices with high pLDDT scores, which can often mislead users regarding their existence and stability. In reality, these regions are frequently representing intrinsically disordered regions. I recommend re-analyzing the isoform models with this additional constraint in mind and advising readers that such models may have limited reliability.

Second, it would be helpful to provide a detailed description of a few structural models of isoforms that exhibit substantial sequence differences from the reference protein over extended regions, where the authors are confident in the accuracy of the isoform models. For these cases, the manuscript should include clear comparative Figures of the two structures, along with the atomic coordinates of the isoform models provided in the Supplementary Data.

Reviewer 2

The authors used Alphafold2 protein structure prediction tool to predict the structure of splice variants. The data set produced is complementary to the Alphafold Protein Structure Database (AFDB) and CHES datasets produced by Sommers et al. and addresses important scientific questions unanswered by either resource.

Understanding the structural implications of different isoforms of human proteins is important due to the implication in diseases. The authors find trends one would expect in comparing homologous proteins, that structural similarity between isoforms correlates with sequence identity, but also some general trends relating to biophysical properties or function when exons are skipped or when alternate exons are used.

The protocols described and metrics used to compare the structures (r^2 -score, SS, surface charge, radius of gyration, and PTM site) are appropriate if the authors are comparing experimentally determined structures of different isoforms. However the structures generated by AF2 are not of sufficient quality to draw the conclusions the authors make particularly with regard to radius of gyration.

In many of the examples shown in figures 1-6 as well as S5 the isoform structures look unfolded and destabilized. It is not clear from the figures if these are outliers or low confidence predictions. For

example in figure 5D, one cannot imagine that the structure shown for the isoform is one in which the protein spends an appreciable amount of time. It is hard to judge if both the general and specific conclusions the authors draw from these analyses are reflective of or simply the pathologies of lower confidence AlphaFold 2 predictions. I would be more inclined to believe the results if the authors used only the highest confidence predictions and devised a method to account for the long regions of loop in their predictions in the metrics they describe.

Specifics

2.1

P6L9, non-AS, AS, replaced or missing should be defined more explicitly in the main body rather than the figure 2 legend.

The introduction states that there are on the order of 100K isoforms but that the authors are only examining 11K after applying the filters for sequence length, why is there such a large reduction?

2.2

A schematic of the 9 different alternative splicing classes would be useful in the main text rather than the supplement

For readers that are not experts in exon selection in alternative splicing and explanation of why the additional breakdown for first and last exons would be helpful

Figure 3C it is not immediately obvious that the plot in the center is the outlined area of 3B and it is not stated in the legend

2.3

For low confidence segments of a prediction, AF2 regularly produces long unstructured regions that are pushed away from the more globular higher confidence regions of the prediction; this could have a large effect on the radius of gyration. Examples are clearly show unstructured interdomain regions and termini that would allow for flexibility and therefore using a single pose for comparison is inappropriate

2.4

Labels in subplots of figure 6D are incredibly tiny

2.5

Labels of axes in figure 7A are incredibly tiny

Figure 7B and 7C could use titles on the plots beyond what is described in the legend

2.6

Labels of axes in figure 8A and 8B are incredibly tiny

Authors' response to reviewers

*We thank the reviewers for their helpful and constructive comments, which have significantly improved the manuscript. We have marked reviewer comments in **bold** and author responses in italics throughout.*

Reviewer #1:

In the manuscript "Predicting the Structural Impact of Human Alternative Splicing" by Song et al., the authors present a systematic analysis of AlphaFold models of protein isoforms generated through alternative splicing. This is a significant area of research that has become feasible on a large scale due to advancements provided by the AlphaFold program.

The structural models predicted by AlphaFold form the foundation of this research; therefore, their quality and robustness must be assured. The authors should make additional efforts to convince readers of the models' accuracy. Specifically, I highlight two issues for consideration.

1. First, in several figures throughout the manuscript, the structures of certain isoforms contain alpha-helices that lack extensive interactions with the rest of the structure. Such occurrences are extremely rare in experimentally determined 3D structures (almost non-existent), yet AlphaFold predicts these isolated alpha-helices with high pLDDT scores, which can often mislead users regarding their existence and stability. In reality, these regions are frequently representing intrinsically disordered regions. I recommend re-analyzing the isoform models with this additional constraint in mind and advising readers that such models may have limited reliability.

We agree that the quality and robustness of the predicted models is very important, especially as this is principally a computational study. Numerous studies have demonstrated that not only are AlphaFold2 predictions of remarkable quality, but also the calibration of the predicted accuracy defined by the average pLDDT is high. Additionally, while many structures predicted by AlphaFold2 may not resemble experimentally characterized structures in both the details and overall topology, in some cases these are due to limitations of the experimental data¹. Nonetheless, we agree that additional analysis will help assess overall and specific predicted accuracy. Regarding the extended alpha helices and intrinsically disordered regions (IDRs), we have systematically analyzed IDRs and identified alternative examples to showcase.

*We included the analysis of intrinsically disordered regions (IDRs) using a state of the art structure based method derived from AlphaFold2 (window-averaged pLDDT scores) and a complementary state of the art sequence based method (SPOT-disorder)². To compare between reference and isoform and between structural and sequence methods, we computed the IDR percentages and described them in the main text and summarized them in **Table S4** and **Figure S6**. Briefly, the fraction of residues called as IDRs are similar between structure and sequence methods, and there is a high consistency for which residues are called IDR, suggesting AlphaFold2 is not broadly over confident in calling disordered regions as ordered. We observed*

a small but significant increase in the IDR percentage in isoforms, consistent with prior results showing an increase of IDRs in alternative splicing proteins³.

*Since IDRs may obscure differences in analysis of the well-structured protein regions, such as radius of gyration, we generated a non-IDR dataset based on the structure-based method. We reanalyzed the radius of gyration results using this dataset and found consistent conclusions: reference proteins generally exhibit higher radius of gyration. For examples of both positive and negative outliers, we exclusively selected cases from the non-IDR dataset. These updated examples are shown in **Figure 5**. Additionally, we performed regression analysis of the per-residue effect using the non-IDR dataset, as shown in **Figure 6B**. Details of the IDR calculation are provided in Methods Section 3.4: Intrinsically Disordered Regions (IDRs) Analysis.*

Here are the figures we updated based on this analysis.

Figure 5:

Figure 6:

tion(A)

Figure S6:

Second, it would be helpful to provide a detailed description of a few structural models of isoforms that exhibit substantial sequence differences from the reference protein over extended regions, where the authors are confident in the accuracy of the isoform models. For these cases, the manuscript should include clear comparative Figures of the two structures, along with the atomic coordinates of the isoform models provided in the Supplementary Data.

*Though we provided examples of high sequence identity but low TM-score, examples with low sequence identity could also be interesting. We added the examples of SRP9 and EEF1AKMT3 in **Figure S3**. They all have sequence identity less than 50% and confident prediction quality ($pLDDT \geq 70$), lack isolated helices, and seem to be well-folded overall.*

A PyMOL session file (.pse) is included in the supplementary materials, containing the atomic coordinates for both the reference and isoform structures. Additionally, we have described these examples in detail in the main text.

Here is the **Figure S3**:

A

SRP9

B

EEF1AKMT3

Reviewer #2:

The authors used Alphafold2 protein structure prediction tool to predict the structure of splice variants. The data set produced is complementary to the Alphafold Protein Structure Database (AFDB) and CHESSE datasets produced by Sommers et al. and addresses important scientific questions unanswered by either resource.

Understanding the structural implications of different isoforms of human proteins is important due to the implication in diseases. The authors find trends one would expect in comparing homologous proteins, that structural similarity between isoforms correlates with sequence identity, but also some general trends relating to biophysical properties or function when exons are skipped or when alternate exons are used.

1. The protocols described and metrics used to compare the structures (T^m -score, SS, surface charge, radius of gyration, and PTM site) are appropriate if the authors are comparing experimentally determined structures of different isoforms. However the structures generated by AF2 are not of sufficient quality to draw the conclusions the authors make particularly with regard to radius of gyration.

In many of the examples shown in figures 1-6 as well as S5 the isoform structures look unfolded and destabilized. It is not clear from the figures if these are outliers or low confidence predictions. For example in figure 5D, one cannot imagine that the structure shown for the isoform is one in which the protein spends an appreciable amount of time. It is hard to judge if both the general and specific conclusions the authors draw from these analyses are reflective of or simply the pathologies of lower confidence Alphafold 2 predictions. I would be more inclined to believe the results if the authors used only the highest confidence predictions and devised a method to account for the long regions of loop in their predictions in the metrics they describe.

Though we used a single pLDDT score for each protein as a measurement to select the predicted structures for metric analysis, different parts of the same proteins could suffer from low prediction quality, and some of them will represent intrinsically disordered regions (IDRs), usually presented as long loops, others could result from the low quality prediction from AlphaFold2. Therefore, we followed your suggestion to use stricter criteria for choosing structures to include in our analyses of various metrics. We reran these analyses in two new ways: (1) filtering by IDR percentage and (2) filtering by overall pLDDT. In addition, we manually curated the examples shown in the figures to ensure that they cannot be explained away by the presence of IDRs or large unstructured regions.

Firstly, we included the intrinsically disordered regions (IDRs) into our analysis, using a structure based method derived from AlphaFold2 (window-averaged pLDDT scores) and a complementary sequence based method (SPOT-disorder)², as described in the Methods section. We described the IDR percentage from both methods in main text and summarized in

Table S4 and Figure S6. Overall, the results from both methods are consistent with each other, and IDR percentage is slightly higher in isoforms (**Figure S6**).

Specifically, for metrics such as radius of gyration, which could be easily affected by IDRs, we created a non-IDR dataset using the results from the structure based method, which excludes IDRs in both isoform and reference structures. The results from the radius of gyration analysis are presented in **Figure 5**. We also reselected examples of positive and negative outliers for the radius of gyration to ensure the chosen examples represent well-folded structures.

Here is the **Figure S6**:

Here is the updated **Figure 5**:

DIPK2A (Radius difference: -8.21A)

FTO (Radius difference: -9.51A)

Secondly, we reanalyze our data using only the high prediction quality ($pLDDT \geq 90$) dataset for different metrics, and the results are mostly consistent with what we get using high and confident dataset.

TM-score:

For TM-score, in panel A, all the nine alternative splicing types are negatively correlated with alternative splicing length, and especially for the alternative splicing types in the begin and end (AFE, ALE, MXE-AFE, and MXE-ALE). Also in panel B, the sequence identity is also strongly positively correlated with TM-score, both results are consistent with what described in **Figure 3** in the main text.

Here is the figure for TM-score based on high prediction quality dataset:

Surface charge:

Using only the structures with high predicted quality, in panel A, surface charge is a little lower in the isoform with median of 0 and in reference the median is 0.99 (p -value: 0.037 Mann–Whitney U test). And correspondingly, in panel B, the difference of surface charge is negatively-skewed, with a median of -1.0. The top five most frequent alternative splicing types among the outliers are AFE, ES, MXE-ALE, ALE, and MXE-AFE (panel C). The overall distribution differs slightly when compared to the high-confidence dataset shown in **Figure 4**. In **Figure 4**, isoforms and references exhibit similar surface charge distributions, with a median difference of 0. Nonetheless, the surface charge distributions in both datasets are comparable and feature significant outliers.

Here is the figure for surface charge based on high prediction quality dataset:

Radius of gyration:

For the radius of gyration, first we updated our results using only the non-IDR dataset, which contains 1,770 isoforms and the results are shown in **Figure 5**. We also analyzed the radius of gyration using only the structures with high prediction quality. The overall distribution is higher in the reference (panel A), which is consistent with the results from both the non-IDR dataset and high and confident dataset, and the difference of radius of gyration is negatively skewed (panel B). There are more negative outliers, and the top top five most frequent alternative splicing types among the outliers are ES, AFE, MXE-ALE, ALE, and ADS (panel C). Here is the figure for radius of gyration based on high prediction quality dataset:

Secondary structure percentage:

The difference is also not significant for secondary structure percentage in isoform and reference using the high prediction quality dataset (panel A).

Here is the figure for secondary structure percentage based on high prediction quality dataset:

Specifics:

2.1

P6L9, non-AS, AS, replaced or missing should be defined more explicitly in the main body rather than the figure 2 legend.

We moved the description of non-AS, AS, replaced or missing from the Figure 2 legend to the main text.

The introduction states that there are on the order of 100K isoforms but that the authors are only examining 11K after applying the filters for sequence length, why is there such a large reduction?

The accurate number of human proteins (including isoforms) is still unclear, but alternative splicing definitely contributes to the variation of proteins. In the reference we mentioned in the manuscript⁴, it is estimated that there are about 100K alternative splicing events in the human body. We previously incorrectly referenced this number as 100K proteins, but have now corrected this in the manuscript.

We only used proteins manually reviewed from SwissProt, and selected proteins for which the “reference” sequence had a maximum of 600 amino acids, which decreased the number of available protein isoforms to 11K.

2.2

A schematic of the 9 different alternative splicing classes would be useful in the main text rather than the supplement

For readers that are not experts in exon selection in alternative splicing and explanation of why the additional breakdown for first and last exons would be helpful

*We agree that the definition of 9 alternative splicing types is important since it relates to our structural metric analysis. Therefore, we added the schematic plots for the 9 different alternative splicing types into **Figure 1**, and also described them in the main text. In Methods part 3.2, we mentioned about why we do additional breakdown for the first and last exons: “Given that a prior study observed that alternative first exons can display mutually exclusive behavior, we introduced the definitions of mutually exclusive exon-alternative first exon (MXE-AFE) and mutually exclusive exon-alternative last exon (MXE-ALE) to differentiate them from alternative first exon (AFE) and alternative last exon (ALE).”*

*Here is the updated **Figure 1**:*

(A) Compare sequence and predict structure

1. Define AS type:

2. Predict isoform structure:

(B) Compare structure

(C) Map expression and predict function

Figure 3C it is not immediately obvious that the plot in the center is the outlined area of 3B and it is not stated in the legend

We added lines between Figure 3B and Figure 3C, which outlined the area of 3B is the center plot of Figure 3C. We also modified the legend of Figure 3C: “The middle box plot is the zoom in plot for the box in panel B.”

Here is the updated **Figure 3**:

2.3

For low confidence segments of a prediction, AF2 regularly produces long unstructured regions that are pushed away from the more globular higher confidence regions of the prediction; this could have a large effect on the radius of gyration. Examples are clearly

show unstructured interdomain regions and termini that would allow for flexibility and therefore using a single pose for comparison is inappropriate.

*The reviewer is correct that single conformation from the AlphaFold2 prediction may not be representative. One possible solution is molecular dynamics (MD) simulation, especially for structures with long unstructured regions². However, those MD simulations are computationally expensive, and thus not feasible at the scale of this study. Thus, we chose instead to focus on reducing the effect of unstructured regions in our analyses. Therefore, we repeated the radius of gyration analyses using the non-IDR subset of our data and updated our results and examples in **Figure 5**. The examples chosen in **Figure 5** are well-folded. Though different conformations of some examples could have different radius of gyration, in general it seems that the examples we chose have clear differences in radius that likely persist across conformations.*

Labels in subplots of figure 6D are incredibly tiny

We remade the labels in this figure (now Figure 7D due to the addition of Fig. 6).
Here is the updated **Figure 7**:

2.5

Labels of axes in figure 7A are incredibly tiny

Figure 7B and 7C could use titles on the plots beyond what is described in the legend

As we added a new main figure, this review refers to Figure 8. We subsetted the original plot of Figure 8A to only show the cell type with average expression larger than 0.4. And we also added titles for Figure 8B and 8C.

Here is the updated **Figure 8**:

2.6

Labels of axes in figure 8A and 8B are incredibly tiny

We remade the labels of axes in this figure (now Figure 9A and 9B due to the addition of Fig. 6), and exchanged the order of Figure 9A and Figure 9B.

Here is the updated **Figure 9**:

D

1. Agarwal, V. & McShan, A. C. The power and pitfalls of AlphaFold2 for structure prediction beyond rigid globular proteins. *Nat. Chem. Biol.* **20**, 950–959 (2024).
2. Tesei, G. *et al.* Conformational ensembles of the human intrinsically disordered proteome. *Nature* **626**, 897–904 (2024).
3. Romero, P. R. *et al.* Alternative splicing in concert with protein intrinsic disorder enables increased functional diversity in multicellular organisms. *Proc. Natl. Acad. Sci. U. S. A.* **103**, 8390–8395 (2006).
4. Pan, Q., Shai, O., Lee, L. J., Frey, B. J. & Blencowe, B. J. Deep surveying of alternative splicing complexity in the human transcriptome by high-throughput sequencing. *Nat. Genet.* **40**, 1413–1415 (2008).

Second round of review

Reviewer 1

I would like to revisit my first comment from the previous report.

In that comment, I noted that isolated alpha-helices (those without contact with the rest of the protein structure) are virtually non-existent in experimentally determined protein structures. I used the term "almost non-existent" because, in about 1% of cases, such helices can be observed only when one terminus is embedded in the structure while the remaining part protrudes from the globular core.

However, AlphaFold predicts a significant number of these isolated alpha-helices and assigns them high pLDDT scores. This is one of the limitations of AlphaFold.

In response to my critique, the authors have not adequately addressed this issue. I continue to observe such cases in Figures 1, 3, and 4. It is essential to identify and exclude these helices from the analysis.

While I understand that this may require substantial revisions to the manuscript, it is a necessary step if the authors wish to make their analysis credible in the eyes of structural biologists.

Authors' response to reviewers

*We thank the reviewer for the constructive comments, which have significantly improved the manuscript. We have marked reviewer comments in **bold** and author responses in italics throughout.*

Reviewer #1:

I would like to revisit my first comment from the previous report.

In that comment, I noted that isolated alpha-helices (those without contact with the rest of the protein structure) are virtually non-existent in experimentally determined protein structures. I used the term "almost non-existent" because, in about 1% of cases, such helices can be observed only when one terminus is embedded in the structure while the remaining part protrudes from the globular core.

However, AlphaFold predicts a significant number of these isolated alpha-helices and assigns them high pLDDT scores. This is one of the limitations of AlphaFold.

In response to my critique, the authors have not adequately addressed this issue. I continue to observe such cases in Figures 1, 3, and 4. It is essential to identify and exclude these helices from the analysis.

While I understand that this may require substantial revisions to the manuscript, it is a necessary step if the authors wish to make their analysis credible in the eyes of structural biologists.

We agree that isolated alpha-helices confidently predicted by AlphaFold2 are problematic. To address this issue, we implemented a rule-based filter to exclude structures containing isolated α -helices, removed these structures from the examples shown in the main figures, and repeated our analyses on the filtered dataset.

*Because isolated helices are solvent-exposed and lack extensive interactions with other regions of the protein structure, we identified them by combining relative solvent accessibility (RSA) with residue-level secondary structure annotations. Specifically, we defined isolated helices as continuous regions of at least 20 exposed helical residues. We removed any structure containing an isolated helix. After applying this filter, we obtained a dataset free of both IDRs and isolated helices. We recalculated the TM-score, surface charge, and radius of gyration analyses on the filtered set of structures and summarized these new results in **Figure S4** and **Figure 6B**. The overall conclusions from these analyses remain consistent whether we retain or remove structures with isolated helices, as described in Results Section 2.2 and 2.3. The filtering criteria are detailed in Methods Section 3.5. We used these filtering criteria to*

update the example structures shown in **Figures 1, 3, 4, 5, 8 and 9**, excluding structures containing isolated helices from these figures.

In total, we now perform our analyses on three separate datasets: (1) all high-confidence structures ($pLDDT \geq 70$), (2) high-confidence structures without extended IDRs, and (3) high-confidence structures without either extended IDRs or isolated helices.

Here are the figures we updated based on this analysis.

Figure 1:

(A) Compare sequences and predict structures

1. Define types:

2. Predict structures:

(B) Compare structures

(C) Map expression and predict function

Figure 1: Workflow for exploring the structural effects of alternative splicing. The workflow is composed of three parts: **(A)** Compare sequence and predict structure: we use AlphaFold2 to predict the structure of 11,161 human alternative spliced isoforms from 5,966 genes. The predicted structures for the “reference” isoform of each gene are publicly available. For each isoform, we annotate the alternative splicing type based on the pattern of spliced exons relative to the reference isoform, and in total we identify nine alternative splicing types among 7,923 isoforms. **(B)** Compare structure: we compare the structures of isoforms for each gene using five different metrics. We calculate template-matching score (TM-score), secondary structure percentage, surface charge, radius of gyration, solvent-accessible area of post translational modification (PTM) sites and **intrinsic disorder region (IDR) analysis**. **(C)** Map expression and predict function: We quantify the expression for each isoform from the Tabula Sapiens scRNA-seq dataset using Kallisto and identify isoform expression differences across human cell types. We use COFACTOR to predict protein functions based on their structures and compare the predicted gene ontology (GO) terms for reference and isoform.

Figure 3:

Figure 3: Structural similarity of isoforms generally reflects sequence similarity, but some structures differ despite similar sequences. (A) Scatter plot of length of alternative splicing region (x-axis) vs. TM-score between reference and alternate isoform structure (y-axis) colored by alternative splicing type, fitted lines are presented for each alternative splicing type. **The same analysis was performed on the non-IDR, non-isolated helices dataset, as referred to in Figure S4A.** (B) Scatter plot of percent sequence identity between reference and alternate isoform (x-axis) vs. TM-score (y-axis).

(PCC=0.565). The high sequence identity (>70%) but low TM-score (<0.5) isoforms are shown in the red box. The same analysis for the CHES dataset and the non-IDRs, non-isolated helices datasets are shown in **Figure S2B** and **Figure S4B**. Examples of isoforms with high sequence identity but low TM-score: CFHR3 (**C**) and MDH2 (**D**). The reference and isoform structures are colored in wheat and pale cyan, and the alternative splicing regions are colored in red. The contact maps for references and isoforms are aligned to the same length. The isoforms predominantly influenced by structural irregularities in loop regions are illustrated in **Figure S7**. The observed variability in the loop regions may arise either from inherent structural disorder or as a consequence of AlphaFold2 predictions characterized by lower accuracy.

Figure 4:

Figure 4: Alternative splicing changes surface charge. (A) Overall surface charge distribution between reference and isoform structures for **SwissProt dataset with high and confident prediction quality** (p-value: 0.304, Mann–Whitney U test), with outliers (defined as $Q1 - 1.5 \text{ IQR}$ and $Q3 + 1.5 \text{ IQR}$) occurring at both ends, $N=7024$. Surface charge distribution for the **CHES dataset and non-IDR, non-isolated helices dataset** are shown in **Figure S2C** and **Figure S4C**. (B) Differences of surface charge for the SwissProt dataset **with high and confident prediction quality**. Difference of surface charge is calculated by using the

isoform surface charge minus the reference surface charge, N=4952. The difference of surface charge for the CHES dataset and the non-IDR, non-isolated helices dataset are shown in **Figure S2D** and **Figure S4D**. **(C)** The five most frequent alternative splicing events in the positive and negative surface charge outliers. The results for the CHES and the non-IDR, non-isolated helices datasets are presented in **Figure S2E** and **Figure S4E**. **(D)** Example of positive surface charge outliers: CXCR3. The reference isoform binds different ligands (CXCL9, CXCL10 and CXCL11) compared to the ligand that binds the alternative isoform (CXCL4). The three aspartic acid residues are labeled in pink. Alternatively spliced portions of the structure are colored red. **(E)** Example of negative surface charge outliers: RCL1, acidic amino acids are labeled in pink sticks.

Figure 5: Alternative splicing changes protein compactness as measured by radius of gyration. (A) Box plot of radius of gyration values for reference and alternate isoform structures (p-value: 1.623e-5, Mann-Whitney U test), N=1770. The median difference between reference and alternate isoform radius is -0.9 Å, the outliers defined as Q1 - 1.5 IQR and Q3 + 1.5 IQR. Radius of gyration distribution for the CHES and the non-IDR, non-isolated helices datasets are shown in **Figure S2F** and **Figure S4F**. **(B)** Differences in radius of gyration for the SwissProt dataset, N=1328. Results for the CHES and the non-IDR, non-

isolated helices dataset are shown in **Figure S2G and Figure S4G**. **(C)** The five most frequent alternative splicing events in positive and negative outliers. Results for the CHES and the non-IDR, non-isolated helices dataset are shown in **Figure S2H and Figure S4H**. **(D)** Examples of positive outliers of radius of gyration (NME2 and **CPPED1**). Alternatively spliced portions of the structure are colored red. **(E)** Examples of negative outliers of radius of gyration (DIPK2A and FTO).

Figure 6: Per-residue effects of alternative splicing types on structural metrics. Per-residue effect analysis on our SwissProt dataset with high and confident prediction quality (**A**) and on the non-IDR, non-isolated helices dataset (**B**). Effect size is colored from green to pink to represent the negative to positive per-residue effect. The size of each dot represents the negative log₁₀ p-value, and dots with insignificant p-value ($\text{p-value} > 0.05$) are colored black. P-value for each effect size is obtained from the `lm` function in R.

Figure 8: Cell-type-specific expression of alternate isoforms. (A) Expression profile for differential expressed reference/isoform across cell types. 'R' in the label stands for each reference and 'I' stands for the alternate isoform. (B) UMAP plot of the cell types, with differentially expressed spliced isoforms of gene KDELR1. (C) and (D) Expression of KDELR1 reference and isoform across eleven cell types. (E) and (F) Structure of KDELR1 reference and isoform. The alternatively spliced region is colored in red

Figure 9: Structure-based prediction of functional changes induced by alternative splicing. Heatmap plot for GO terms frequently lost (A) or gained (B) across nine alternative splicing types, colored by p-value. The text colors for GO terms indicate which sub-ontology they come from. (C) Structures for EPHX2 reference and EPHX2 isoform 2 and predicted GO terms lost by EPHX2 isoform 2. The alternative splicing regions are colored in red. The lipid phosphatase domain is circled in the structure. (D) Structures for TFAZZIN reference and TFAZZIN isoform 7 and predicted GO terms lost by TFAZZIN isoform 7. The alternative splicing regions are colored in red. The structures encoded by exon 5 and 7 are circled.

Figure S4: Structural metrics analysis for the non-IDR, non-isolated helices dataset. (A) Scatter plot of length of alternative splicing region (x-axis) vs. TM-score for 999 alternate isoforms with their reference structures (y-axis) colored by alternative splicing type, fitted lines are presented for each alternative splicing type. (B) Scatter plot of percent sequence identity between reference and alternate isoform (x-axis) vs. TM score (y-axis) (PCC=0.798, N=1,359). (C) Overall surface charge distribution between reference and isoform structures (p-value: 0.58, Mann-Whitney U test, N=1,359). (D) Differences of surface charge from the non-IDR, non-isolated helices dataset, N=1,012. (E) The five most frequent alternative splicing events in the positive and negative surface charge outliers. (F) Overall radius of gyration distribution between reference and isoform structures (p-value: 9.497e-9, Mann-Whitney U test, N=1,359). (G) Differences of radius of gyration from the non-IDR, non-isolated helices dataset, N=1,012. (H) The five most frequent alternative splicing events in the positive and negative radius of gyration outliers.

Third round of review

Reviewer 1

The authors responded to my comment.